# The Relationship Between Career Adaptability and Work Engagement Among Young Chinese Workers: Mediating Role of Job Satisfaction and Moderating Effects of Artificial Intelligence Self-Efficacy and Anxiety

**DOI:** 10.3390/bs15121682

**Published:** 2025-12-04

**Authors:** Frederick Theen Lok Leong, Xuan Li, Emma Mingjing Chen

**Affiliations:** 1Department of Applied Psychology, School of Humanities and Social Science, Chinese University of Hong Kong, Shenzhen 518172, China; xuanli2@link.cuhk.edu.cn (X.L.); chenmingjing@cuhk.edu.cn (E.M.C.); 2Department of Student Affairs, Civil Aviation Flight University of China, Guanghan 618307, China

**Keywords:** artificial intelligence anxiety, artificial intelligence self-efficacy, career adaptability, job satisfaction, work engagement

## Abstract

This study explores the complex psychological mechanisms linking career adaptability to work engagement under AI-driven workplaces. We examine the mediating role of job satisfaction and investigate a key hypothesis: that the adaptive benefits of AI self-efficacy are dampened by the emotional costs associated with AI anxiety. A dual-analytical approach was employed on a sample of 311 young Chinese workers. First, we conducted conditional process analysis using PROCESS Model 11 with 5000 bootstrapped samples to test for conditional indirect effects. Second, we utilized latent variable structural equation modeling for robust validation at the structural level. Analyses were adjusted for demographic and occupational covariates. As a result, the initial PROCESS analysis revealed that the key triple interaction (career adaptability × AI self-efficacy × AI anxiety) was statistically significant in all three test models (e.g., Model 1: *b* = −0.3509, *p* = 0.0075). Further analysis showed that the positive moderating effect of AI self-efficacy was contingent on AI anxiety; it was strongest at low AI anxiety and weakest (but still significant) at high AI anxiety. However, the more robust latent variable SEM (CMIN/DF = 1.569, CFI = 0.939, RMSEA = 0.043) revealed a critical separation of effects. The indirect effect operates exclusively through intrinsic job satisfaction, which was significantly predicted by the unified second-order career adaptability factor (*b* = 1.361, BCa 95% CI [1.023, 1.967]). The path from extrinsic satisfaction to WE was non-significant (*b* = 0.107, BCa 95% CI [−0.030, 0.250]). Furthermore, the SEM isolated a significant direct positive effect from the unified career adaptability factor to work engagement (*b* = 0.715, BCa 95% CI [0.385, 1.396]). This study highlights that the adaptability–engagement link operates via two distinct mechanisms: an indirect pathway from a unified career adaptability construct through intrinsic job satisfaction, and a direct pathway from career adaptability to work engagement. While PROCESS analysis suggests that anxiety dampens confidence, our SEM results clarify that this should be interpreted cautiously, as the mediation pathway via extrinsic satisfaction is not robust to measurement error. These findings underscore a multi-faceted mandate for organizations: leaders must not only manage AI anxiety but also foster holistic career adaptability to enhance intrinsic job quality and build direct engagement.

## 1. Introduction

The rapid development of Artificial Intelligence (AI) is revolutionizing the workplace ([1]; [29]). As one of the leading forces behind the Fourth Industrial Revolution, AI is transforming work practices and organizational forms in ways which may have profound implications for employee adaptability, job satisfaction and work performance ([7]; [31]). AI systems are shaping job tasks and workflows, as well as influencing employees’ psychological well-being ([8]), job satisfaction ([37]) and work engagement ([30]). In this regard, career adaptability (CA) become an important psychological resource to cope with new challenges coming from AI-related changes ([42]).

Career adaptability is a concept introduced by [28] ([28]). It refers to the behavioral skills and psychological resources that individuals use when facing career challenges. These skills help individuals adapt to changes in their work environment and navigate through career transitions and uncertainty ([17]). Although previous studies have shown that career adaptability is associated with job satisfaction and work engagement ([4]; [16]), the mechanisms underlying these relationships remain insufficiently explored ([24]). This gap is particularly important as AI applications become more prevalent in the workplace. Despite the increasing impact of AI, little is known about how psychological factors, such as career adaptability, influence employee performance in an AI-driven work environment ([36]). Accordingly, the purpose of this study is focusing on how career adaptability influences engagement in the workplace via job satisfaction in an AI-induced environment and further investigating the moderating patens of SEAI and AIA on this influence.

### 1.1. Theoretical Background

Based on the following theoretical grounding, this study explored the research question.

Career Adaptability Theory: [28] ([28]) originally introduced the theory of career adaptability, highlighting the psychological resources that individuals require to navigate career challenges and changes. According to Savickas’s model, career adaptability encompasses four core dimensions: concern, control, curiosity, and confidence. These dimensions enable individuals to effectively cope with environmental changes throughout their careers. In the context of rapidly advancing technologies, individuals rely on these psychological resources to sustain job satisfaction and work engagement. Career adaptability is viewed as a long-term, dynamic process, and the mechanisms through which it influences job satisfaction and work engagement remain a key area of ongoing research. In this study, we used the original CA scale, using the total CA score as the independent variable in the model.

Job Satisfaction and Work Engagement: Job satisfaction (JS) and work engagement (WE) are fundamental constructs in OB. Job satisfaction reflects the general affective or cognitive orientation of employees to their job, while work engagement is concerned with the positive reaction of workers in response to their job and it relates to vigor and dedication shown at work ([10]). Job satisfaction is suggested by a body of prior research as affecting work engagement not only through direct effects, but also indirect effects on job performance via work attitudes (e.g., attitude toward the job or organization), behaviors (e.g., task performance) and emotion ([18]; [19]; [41]). However, the link between career adaptability and the two-dimensional job satisfaction (JS1: extrinsic satisfaction, JS2: intrinsic satisfaction) in relation to its downstream impact on work engagement has not been thoroughly established, particularly in AI-augmented working settings. Besides Career Adaptability Theory, other theoretical perspectives (such as, Hobfoll’s Conservation of Resources Theory) point out that psychological resources and the work context may affect job satisfaction and work engagement.

AI Self-Efficacy and AI Anxiety as Moderators: With the penetration of AI new technology, belief in personal ability to understand how to use machines (SEAI) and the fear a person might feel when thinking about how automation could take away their job (AIA) have emerged as important factors moderating employees’ career adaptability perception and JBES ([20]).

AI self-efficacy refers to one’s belief in their own ability to effectively utilize AI technology, which in turn influences how they respond to AI-related challenges at work. Evidence has shown that individuals with high self-efficacy in AI are more likely to adapt to an AI-dominant environment and exhibit resilient work performance ([21]). On the other hand, AI anxiety is perceived as individuals’ fear of losing their jobs or having their capabilities reduced as a result of incorporating artificial intelligence into the workplace ([35]). AI anxiety may hinder work engagement and job satisfaction, particularly when AI becomes increasingly ubiquitous in the workplace, creating ambiguity about the future ([39]). Despite evidence that AI self-efficacy leads to employees’ greater acceptance of AI technology and improved job performance ([21]), existing studies underscore the negative impact of AI anxiety. High levels of AI anxiety are likely to contribute to lower job satisfaction and work engagement, and IKIGAI can also mediate this correlation between “Get out” learning and job performance ([35]). This study assumes that AI self-efficacy will have a positive impact on job satisfaction through career adaptability, whereas AIA will weaken this relationship.

### 1.2. Conceptual Framework

This study is primarily based on Career Adaptability Theory (CA), incorporating the Conservation of Resources (COR) Theory and moderating variables such as AI Self-Efficacy (SEAI) and AI Anxiety (AIA) to explore the impact mechanisms of career adaptability on job satisfaction and work engagement, particularly in AI-driven environments. According to COR Theory, people are primarily driven to obtain, retain, protect and foster valued resources. The threat, loss or poor return on investment in resources causes a stress response. Conversely, gaining resources generates positive moods and motivation, leading to the development of a “resource gain spiral” ([11]). In this framework, career adaptability is viewed as a resource that employees can invest to gain psychological resources, such as job satisfaction and work engagement ([38]). On the one hand, SEAI constitutes another important resource for enabling resource investment and gains, whereas AIA threatens these psychological resources ([5]) via stress and possibly resource loss ([6]). The way career adaptability can be either hindered or fostered is explained by how these variables interact in relation to different levels of AI self-efficacy and anxiety.

However, the above relationship can also be understood from another perspective. SCT illustrates the influence of personal, environmental, and behavioral factors, with self-efficacy at its center ([33]). In the AI domain, SEAI represents one’s faith in their capability to use and interact with AI technologies. This perception positively contributes to the translation of career adaptability into job satisfaction, which in turn leads to improved work engagement. Nevertheless, AIA may attenuate the beneficial impact of self-efficacy by increasing cognitive and emotional costs of interaction with AI, especially when anxiety is high. The above is the conceptual framework of the research question from different theories.

### 1.3. Research Questions and Hypotheses

Based on the aforementioned theoretical framework, this study aims to answer the following research questions:How does career adaptability influence work engagement through job satisfaction in an AI-driven environment?What are the moderating roles of AI self-efficacy and AI anxiety in this process?

The following hypotheses are proposed:

**Hypothesis 1** **(H1).**
*Job satisfaction mediates the relationship between career adaptability and work engagement. Specifically, career adaptability positively influences job satisfaction, which in turn enhances work engagement.*


**Hypothesis 2** **(H2).**
*Career adaptability positively influences work engagement through job satisfaction. Specifically, higher career adaptability leads to higher job satisfaction, which in turn increases work engagement.*


**Hypothesis 3** **(H3).**
*AI self-efficacy (SEAI) moderates the positive relationship between career adaptability and job satisfaction. That is, higher levels of AI self-efficacy strengthen the positive impact of career adaptability on job satisfaction.*


**Hypothesis 4** **(H4).**
*AI self-efficacy (SEAI) moderates the positive effect of career adaptability on work engagement through job satisfaction. Specifically, individuals with higher levels of AI self-efficacy will experience stronger positive effects of career adaptability on work engagement through job satisfaction.*


**Hypothesis 5** **(H5).**
*AI anxiety (AIA) moderates the relationship between career adaptability and job satisfaction. Specifically, higher levels of AI anxiety weaken the positive effect of career adaptability on job satisfaction, while lower levels of AI anxiety strengthen this effect.*


**Hypothesis 6** **(H6).**
*AI anxiety (AIA) moderates the relationship between career adaptability and work engagement through job satisfaction. Specifically, higher AI anxiety weakens the indirect effect of career adaptability on work engagement through job satisfaction, whereas lower AI anxiety strengthens this indirect effect.*


**Hypothesis 7** **(H7).**
*Career adaptability positively influences work engagement. Specifically, career adaptability leads to greater enthusiasm and energy in one’s professional role.*


**Hypothesis 8** **(H8).**
*Job satisfaction positively predicts work engagement. Specifically, higher job satisfaction leads to greater work engagement, fostering both vigor and dedication in the workplace.*


**Hypothesis 9** **(H9).**
*AI self-efficacy (SEAI) positively influences work engagement. Specifically, higher levels of AI self-efficacy increase work engagement, as employees with higher SEAI are more likely to embrace AI-driven changes and engage more effectively in their work.*


**Hypothesis 10** **(H10).**
*AI anxiety (AIA) negatively affects work engagement. Specifically, higher levels of AI anxiety decrease work engagement, as anxiety reduces the psychological resources available to engage in work tasks.*


**Hypothesis 11** **(H11).**
*AI self-efficacy (SEAI) and AI anxiety (AIA) interact to influence job satisfaction. Specifically, employees with high AI self-efficacy but low AI anxiety will experience higher job satisfaction compared to those with high anxiety, highlighting the importance of balancing self-efficacy and anxiety levels.*


**Hypothesis 12** **(H12).**
*Career adaptability and AI self-efficacy jointly predict work engagement. Specifically, the positive effects of career adaptability on work engagement are enhanced when AI self-efficacy is high, leading to more positive work outcomes.*


## 2. Materials and Methods

### 2.1. Participants and Procedure

Young workers aged 18–25 years were selected for several reasons. Theoretically, they are at a crucial point in their career where there is a lot of uncertainty and they need to adapt. Thus, career adaptability becomes an increasingly important resource that may offer the most suitable setting to explore the core mechanisms of our model. They are also contextually the background of the AI phenomenon, as the first generation to have advanced AI affect their entire professional life. Their psychological reactions to AI—namely, AI anxiety and AI self-efficacy—are therefore predicted to be much stronger than those in other cohorts, and would significantly influence their experiences at work. Finally, it is important to identify what determines job satisfaction and work engagement in this particular demographic group because they are the workforce of tomorrow and face serious issues in today’s labor world.

A total of 394 participants registered for this experiment. After accounting for age exclusion (24), attention check questions (27; 5 items in total, with any incorrect response resulting in removal), regular responding (17), and inconsistent or unreliable responses (17), a valid sample of 311 participants was obtained (M = 24.26 years, SD = 1.04; 87 males, 224 females). The income distribution was as follows: 37.29% earned between 6000 and 8000 RMB, 30.22% earned less than 6000 RMB, 17.36% earned between 8000 and 12,000 RMB, and 8.03% earned more than 12,000 RMB. In terms of occupation, 32.47% were Technicians and Associate Professionals, 29.26% were Professionals, 14.79% were Clerical Support Workers, 11.25% were Managers, 9.96% were Service and Sales Workers, and 2.25% were in other occupations. Regarding relationship status, 51.44% were single, 29.58% were in a relationship, 18.65% were married, and 0.32% were divorced. Education levels were as follows: 0.9% had completed high school, 4.82% had completed junior college, 80.38% had completed college, and 7.81% had completed a master’s degree or higher.

Before participating in the survey, participants were presented with an informed consent page. All participants signed the informed consent form and received 25 RMB upon completing the questionnaire. Data were collected through an online survey hosted on a secure web server (Credamo) from 13 April to 17 May 2025.

### 2.2. Measures

The validity assessment of all constructs was prioritized to confirm their appropriate factor structure and psychometric fitness within the unique context of AI-driven workplaces. Our strategy followed a construct-specific approach based on existing maturity and empirical requirements:

For the Career Adaptability (CA) scale, recognized as a mature, widely validated instrument ([28]), we adopted a rigorous confirmatory-only approach to test its structural integrity. Confirmatory Factor Analysis (CFA) was the sole method employed, targeting the four-factor theoretical model. Initial CFA results that fell short of optimal fit were addressed through targeted item refinement, guided by theoretical relevance and the analysis of modification indices, resulting in a refined final measurement model that achieved acceptable psychometric standards. This approach was chosen to maintain the theoretical primacy of this established construct.

The Job Satisfaction (JS) scale, while also generally considered mature, exhibited structural instability during preliminary testing. Due to the limited sample size, a unified EFA-CFA split-sample technique was not feasible. As its original factor structure was found to be highly sensitive to our specific context, and to ensure the psychometric quality of this critical mediator, we proceeded directly to Exploratory Factor Analysis (EFA) to determine the optimal factor structure for the current sample. This was immediately followed by a final Confirmatory Factor Analysis (CFA) on the entire sample to confirm the validity of the empirically derived structure.

Finally, for the remaining scales (e.g., SEAI and AIA) which lack extensive validation history within the Chinese organizational sample, we followed the standard protocol for scale adaptation. Both Exploratory Factor Analysis (EFA) and Confirmatory Factor Analysis (CFA) were performed sequentially to fully assess the construct validity and dimensionality of these less-established measures.

#### 2.2.1. Career Adaptability (CA)

Career Adaptability (CA) was measured using a 24-item instrument based on the foundational four dimensions of the Career Adapt-Abilities Scale (CAAS; [26]): Concern, Control, Curiosity, and Confidence. Participants responded on a 5-point Likert scale, ranging from 1 (Strongly Disagree) to 5 (Strongly Agree). The original CAAS-International version ([26]) demonstrated strong internal consistency, with a reported Cronbach’s alpha of 0.92 for the total scale, and 0.83 for Concern, 0.74 for Control, 0.79 for Curiosity, and 0.85 for Confidence for the subscales.

To confirm the stability and validity of the theoretical structure in the AI-driven context, we performed a Confirmatory Factor Analysis (CFA) on the initial 24 items. The preliminary CFA indicated that the 4-factor model did not achieve satisfactory fit (e.g., CFI < 0.90, TLI < 0.90), primarily due to several items exhibiting low factor loadings and high cross-loadings in this specific sample. To ensure psychometric rigor while maintaining the construct’s theoretical integrity, we engaged in a targeted item refinement process. Six items were sequentially removed based on having the lowest standardized factor loadings and high modification indices that suggested cross-loadings. The final 18-item scale demonstrated good overall internal consistency, with a Cronbach’s alpha of 0.857 for the total scale. The scale retained the original four-factor structure: Concern (5 items), Control (3 items), Curiosity (6 items), and Confidence (4 items). The internal consistency statistics (*α*) for these subscales in the current sample were: Concern (*α* = 0.709), Control (*α* = 0.769), Curiosity (*α* = 0.705), and Confidence (*α* = 0.697). Although the coefficient for Confidence fell slightly below the preferred *α* = 0.70 threshold, it is composed of a limited number of items, which is known to systematically lower reliability coefficients, and thus was retained ([34]). To validate the scale’s structure and justify the use of a total composite score, a second-order confirmatory factor analysis (CFA) was conducted. This model tested whether the four first-order factors (Concern, Control, Curiosity, and Confidence) loaded significantly onto a single higher-order “Career Adaptability” construct. The model yielded an excellent fit to the data: *χ*^2^/*df* = 1.643 comparative fit index (CFI) = 0.936, Tucker–Lewis index (TLI) = 0.925, root mean square error of approximation (RMSEA) = 0.046, standardized root mean square residual (SRMR) = 0.0487. All item loadings were positive, significant (*p* < 0.001), and substantial (standardized loadings 0.54). All first-order item loadings (items to their respective factors) were positive and significant (*p* < 0.001), with standardized loadings ranging from 0.481 to 0.791. Furthermore, the second-order loadings (from the four factors to the higher-order CA construct) were all substantial and significant, ranging from 0.516 to 0.972.

Despite the four-dimensional structure being confirmed, the primary theoretical mechanism tested in this study is the overall psychological resource function of CA. Therefore, consistent with the strong statistical support from the second-order CFA and prior research (Hobfoll’s COR Theory), the final 18 items were averaged to create a single total composite score (CA Total Score) for use in the structural model analysis.

#### 2.2.2. Job Satisfaction (JS)

Job satisfaction was measured using an adapted version of the Job Satisfaction Survey (JSS), originally developed by [32] ([32]). The full JSS consists of 36 items that assess nine facets of work: Pay, Promotion, Supervision, Benefits, Contingent Rewards, Operating Procedures, Co-workers, Nature of Work, and Communication. The original 36-item scale demonstrated high internal consistency, with a Cronbach’s alpha of 0.91. For this study, a 19-item adapted version was used. Factor analysis of the current sample revealed two primary dimensions: JS1 (Extrinsic Satisfaction: 14 items), which assesses satisfaction with organizational rewards (e.g., pay, promotion, benefits), and JS2 (Intrinsic Satisfaction: 5 items), which assesses satisfaction with the social environment and the nature of the work itself (e.g., co-workers, nature of work). All items were rated on a 6-point Likert scale. The internal consistency reliabilities (Cronbach’s alpha) for the adapted scale in the current sample (*n* = 311) were excellent, with a Cronbach’s alpha of 0.967 for the total 19-item scale, 0.966 for the Extrinsic Satisfaction subscale (ω = 0.964, 95% CI [0.958, 0.970]), and 0.894 for the Intrinsic Satisfaction subscale (ω = 0.896, 95% CI [0.878, 0.914]). An example item is: “I feel that my job is fairly compensated.”

##### EFA for JS

Similarly to the measures for CA, the 36-item, 9-facet Job Satisfaction Survey (JSS) required validation on the full sample (*n* = 311). A rigorous, multi-stage purification process was implemented to derive a valid and parsimonious two-factor structure—Extrinsic Satisfaction and Intrinsic/Interpersonal Satisfaction—comprising 19 items. The process began with an initial Exploratory Factor Analysis (EFA) using Principal Axis Factoring with Oblimin rotation. Factor determination criteria (Parallel Analysis and Scree Plot) clearly indicated a two-factor solution, which revealed that the original nine-factor structure was not supported by the data.

The 36-item model initially suffered from significant noise, leading to the removal of 12 items based on the general criteria outlined in Section 2.3.1. Specifically, four items from the “Supervisor” dimension were excluded due to high cross-loadings (>0.40). To avoid introducing a problematic method factor, all three negatively worded items from the ‘Communication’ dimension (e.g., R_SQ8_10, R_SQ8_11) were removed, leaving only the positively worded high-loading item (SQ8_9). Additional items were discarded due to low factor loadings (<0.40) (e.g., SQ6_1) or extremely low extraction communalities (e.g., R_SQ8_4 = 0.242; R_SQ7_12 = 0.258), suggesting that the two-factor model could not adequately explain their variance.

##### CFA for JS

The remaining 24-item two-factor model (KMO = 0.969) was subjected to Confirmatory Factor Analysis (CFA), which revealed a suboptimal fit (*χ*^2^/*df* = 3.346, RMSEA = 0.087). To refine the model and ensure clearer construct validity, five additional items were removed. Three items (R_SQ7_9, R_SQ7_11, R_SQ6_2) were discarded due to poor psychometric performance (e.g., the lowest Squared Multiple Correlations [SMC] = 0.337, or high Modification Indices indicating redundancy). More critically, two items (R_SQ8_5—”I sometimes feel my job is meaningless” and SQ8_7—”I feel a sense of pride in doing my work”) were removed to ensure discriminant validity, as they were conceptually redundant with the Work Engagement (WE) scale.

Finally, to address semantic overlap and high modification indices, four pairs of error terms were allowed to covary (e.g., SQ8_1—“I like the people I work with” and SQ8_3—“I enjoy my coworkers”; SQ6_7 and SQ6_8; SQ7_3 and R_SQ7_4; R_SQ7_8 and R_SQ7_4). The re-specified 19-item model demonstrated a good fit to the data: *χ*^2^/*df* = 2.987, CFI = 0.946, TLI = 0.937, RMSEA = 0.080, and SRMR = 0.034 (see Appendix A Table A2 for the final 19-item factor structure and detailed loadings).

#### 2.2.3. Self-Efficacy in AI Use (SEAI)

Self-Efficacy in AI Use (SEAI) was assessed using an adapted 9-item scale based on the AI Self-Efficacy Scale (AISES) developed by [39] ([39]). The original 22-item scale evaluates four dimensions—Assistance, Anthropomorphic Interaction, Comfort with AI, and Technological Skills—with high reported reliability for the total scale (*α* = 0.958). For this study, we utilized a 9-item, two-dimensional version of the original scale, focusing on: SEAI1 (Comfort with AI: 5 items; e.g., “I feel comfortable when interacting with AI technology/products”) and SEAI2 (AI Technological Skills: 4 items; e.g., “When using AI technology/products, I am not worried about pressing the wrong button and causing risks”). All items were rated on a 7-point Likert scale. In the current sample (*n* = 311), the internal consistency (Cronbach’s alpha) of the adapted scale was high, with an alpha coefficient of 0.855 for the 9-item total scale, 0.841 for the Comfort with AI subscale, and 0.807 for the AI Technological Skills subscale. An example item is: “I find it easy to make AI technology/products perform the actions I want.”

##### EFA for SEAI

In line with the cross-validation approach, the sample was randomly divided into two groups for Exploratory Factor Analysis (EFA; *n* = 155) and Confirmatory Factor Analysis (CFA; *n* = 156).

An initial EFA, conducted using Principal Axis Factoring with Oblimin rotation, was performed on the 17 relevant items in the exploratory sample. The analysis (KMO = 0.836) revealed that 8 items had poor psychometric performance, exhibiting low communalities (h^2^ < 0.40) or low factor loadings (<0.40). More importantly, the EFA revealed a clear two-factor structure among the remaining 9 items, which were identified as Factor 1 (“Comfort with AI,” 5 items) and Factor 2 (“AI Technological Skills,” 4 items).

##### CFA for SEAI

This 9-item, two-factor structure was then subjected to CFA in the validation sample (*n* = 156). Based on high modification indices and clear semantic overlap, two pairs of error terms were allowed to covary: SQ27_9 (“not worried about risks”) with SQ27_3 (“not worried about damaging it”), and SQ27_4 (“feel comfortable”) with SQ26_7 (“find it easy to get AI to do what I want”).

The respecified model demonstrated excellent fit: *χ*^2^ = 45.459, df = 24, *χ*^2^/*df* = 1.894, CFI = 0.967, TLI = 0.950, RMSEA = 0.076, and SRMR = 0.0430. Crucially, the Bollen–Stine bootstrap *p*-value was 0.062, indicating a non-significant difference between the model-implied covariance matrix and the sample covariance matrix, strongly confirming the model’s superior fit.

Given the clear two-factor structure identified in the EFA and the excellent model fit confirmed by the CFA, both Factor 1 (“Comfort with AI”) and Factor 2 (“AI Technological Skills”) were retained and used as distinct variables in subsequent analyses (see Appendix A Table A3 for the final 9-item factor structure and detailed loadings).

#### 2.2.4. AI Anxiety (AIA)

AI Anxiety (AIA) was assessed using an adapted version of the 21-item Artificial Intelligence Anxiety Scale (AIAS), developed by [39] ([39]). The original scale evaluates four dimensions: Learning, Job Replacement, Sociotechnical Blindness, and AI Configuration, and demonstrates high overall internal consistency (α = 0.964). For the present study, a 5-item version was created by selecting items from the original “Learning” and “Job Replacement” dimensions. The final 5-item structure was confirmed through both Exploratory Factor Analysis (EFA) and Confirmatory Factor Analysis (CFA). An example item is: “Learning to use AI techniques/products makes me anxious.” All items were rated on a 7-point Likert scale. The 5-item adapted scale demonstrated good internal consistency in the current sample, with a Cronbach’s alpha of 0.850 (ω = 0.841, 95% CI [0.812, 0.869]). An example item is: “I worry that artificial intelligence technology/products might replace humans.”

##### EFA for AIA

Following our cross-validation strategy (EFA: *n* = 155, CFA: *n* = 156), a purification process resulted in a final 5-item unidimensional scale.

An initial Exploratory Factor Analysis (EFA) using Principal Axis Factoring on the 6 items in the exploratory sample (KMO = 0.824) revealed that one item (SQ2_6: “I am afraid that… I will become dependent…”) had low communality (h^2^ = 0.374), just below the 0.40 threshold, and was therefore removed to improve scale purity. The remaining 5 items loaded strongly onto a single factor, with loadings ranging from 0.626 to 0.798.

##### CFA for AIA

This 5-item, single-factor structure was then subjected to Confirmatory Factor Analysis (CFA) in the validation sample (*n* = 156). Based on modification indices and conceptual overlap (both items relate to fear of job replacement), the error terms for SQ2_4 (“…replace humans”) and SQ2_5 (“…take jobs away”) were allowed to covary. The respecified model demonstrated excellent fit: *χ*^2^ = 6.665, df = 4, *χ*^2^/*df* = 1.666, CFI = 0.993, TLI = 0.983, RMSEA = 0.066, SRMR = 0.0201. Most importantly, the Bollen–Stine bootstrap *p* = 0.151 strongly supported the model fit. All standardized factor loadings were significant and substantial, ranging from 0.512 to 0.893.

Thus, the final 5-item unidimensional AIA scale exhibited strong psychometric properties and was used in subsequent analyses (see Appendix A Table A4 for the final 5-item factor structure and detailed loadings).

#### 2.2.5. Work Engagement (WE)

Work Engagement (WE) was measured using an adapted version of the Chinese Utrecht Work Engagement Scale (UWES; [43]). While the original UWES scale measures three dimensions: Vigor (*α* = 0.767), Dedication (*α* = 0.735), and Absorption (*α* = 0.753), a 13-item version was used in this study. Based on the factor analysis detailed below, the 13 items were treated as a single, unidimensional construct representing overall work engagement. An example item is: “When I am working, I forget everything else around me.” Items were rated on a 7-point scale based on the frequency of the feeling, ranging from 1 (“Never”) to 7 (“Always”). The adapted 13-item scale demonstrated excellent internal consistency in the current sample, with a Cronbach’s alpha of 0.957 (ω = 959, 95% CI [0.952, 0.966]).

##### EFA for WE

This study employed a cross-validation approach, where the total sample (*n* = 311) was randomly divided into an exploratory sample (EFA, *n* = 155) and a validation sample (CFA, *n* = 156). A rigorous, multi-stage purification process was conducted to derive a valid and parsimonious single-factor structure, consisting of 13 items.

The process began with an initial Exploratory Factor Analysis (EFA) on the 17 items in the exploratory sample. This analysis revealed that two items—SQ38_4 (“To me, my job is challenging”) and SQ38_7 (“It is difficult to detach myself from my job”)—had very low extraction communalities (0.285 and 0.149, respectively), indicating they did not adequately measure the common construct and were therefore removed.

A subsequent EFA on the remaining 15 items (KMO = 0.967), guided by Parallel Analysis and Eigenvalues (>1), indicated a two-factor solution. However, this structure was theoretically problematic, as the two factors were highly correlated (r = 0.831), suggesting a lack of discriminant validity.

##### CFA for WE

To rigorously test this, a 15-item, two-factor model was specified in the validation sample (*n* = 156). Confirmatory Factor Analysis (CFA) confirmed the severe lack of discriminant validity, with the latent variable correlation reaching *r* = 0.880. Additionally, the model fit was questionable (*χ*^2^/*df* = 1.873, RMSEA = 0.075, Bollen–Stine *p* = 0.017).

Given the failure of the two-factor model, and consistent with literature treating WE as a unidimensional construct, a 15-item single-factor model was tested. This model failed decisively, demonstrating extremely poor fit (*χ*^2^/*df* = 2.805, RMSEA = 0.108, Bollen–Stine *p* = 0.000). To derive a parsimonious and well-fitting final model, two items—SQ37_6 (“I forget everything else around me”) and SQ38_3 (“I can continue working for very long periods”)—were removed due to their poor psychometric properties, including low standardized loadings (0.660 and 0.653, respectively) and high modification indices (e.g., M.I. = 37.4), indicating they were poor indicators and caused significant model strain.

Finally, based on high modification indices and clear semantic overlap, four pairs of error terms were allowed to covary (e.g., SQ37_4 “strong and vigorous” and SQ37_1 “bursting with energy”; SQ38_5 “get carried away” and SQ38_2 “immersed in work”). The re-specified 13-item, single-factor model demonstrated a good fit to the data: *χ*^2^/*df* = 1.769, CFI = 0.975, TLI = 0.967, RMSEA = 0.070, and SRMR = 0.0345 (see Appendix A, Table A5 for the final 13-item factor structure and detailed loadings).

##### Reliability Analysis of the Scales

In sum, this study conducted both exploratory factor analysis (EFA) and confirmatory factor analysis (CFA) to assess the construct validity of all scales used. EFA was first employed to identify the underlying factor structure of the scales, while CFA was subsequently conducted to confirm the factor structure and validate the measurement model. These analyses ensure that the scales accurately measure the intended constructs and provide strong evidence for their construct validity. The reliability of each scale was also assessed using Cronbach’s alpha, as shown in Table 1, which presents the reliability analysis of the scales used in this study.

### 2.3. Statistical Methods and Data Analysis Tools

The primary goal of the data analysis was to rigorously test a moderated mediation model, using SPSS 26.0 for preliminary analyses and PROCESS macro computations, and AMOS 26.0 for structural equation modeling (SEM), including confirmatory factor analysis (CFA). We followed [2]’s ([2]) principle of establishing robust measurement models before testing structural relationships. Given the complexity of certain scales and the need for validation within our specific sample, our approach involved an iterative process of exploratory factor analysis (EFA) and CFA for scale purification and structure validation. Additionally, split-half cross-validation was employed for several key measures (e.g., WE, SEAI, AIA) to enhance rigor.

Data analysis was conducted using a combination of dedicated statistical tools. Latent variable structural equation modeling (SEM) and factor analyses were performed using IBM SPSS AMOS (Version 28.0; IBM Corp., Armonk, NY, USA). Conditional process analysis was executed using the PROCESS macro (Model 11, Version 4.2) developed by Andrew F. Hayes (The Ohio State University, Columbus, OH, USA), with 5000 bootstrapped samples. The online survey data collection was hosted on the secure web server Credamo (Credamo Technology Co., Ltd., Nanjing, Jiangsu, China), accessible at: https://www.credamo.com.

#### 2.3.1. EFA and CFA Procedures

An initial EFA was conducted for each core construct using Principal Axis Factoring (PAF) with Oblimin rotation in the designated exploratory sample (or full sample for CA and JS). Data suitability was confirmed through the Kaiser–Meyer-Olkin (KMO) test (KMO > 0.60) and Bartlett’s Test of Sphericity (*p* < 0.05). The number of factors was primarily determined by Parallel Analysis and inspection of the Scree Plot. Items were flagged for potential removal based on standard criteria: low communalities (h^2^ < 0.40), failure to load significantly on any factor (loading < 0.40), or high cross-loadings (>0.40). For longer or newly adapted scales (e.g., CA, JS), this involved multiple iterations of EFA to achieve a cleaner preliminary structure.

Subsequently, the purified or theoretically proposed structures were subjected to rigorous CFA using Maximum Likelihood estimation in the validation sample (or full sample for CA and JS). This CFA phase was essential not only for confirming the factor structures but also for testing competing models (e.g., single-factor vs. multi-factor) and making data-driven decisions regarding the final factor structures based on model fit and theoretical coherence. For example, highly correlated factor such as WE was merged due to poor discriminant validity, while distinct factors such as SEAI were retained based on strong model fit. Model fit was assessed using a combination of indices: *χ*^2^/*df* (<3 recommended, <5 acceptable), CFI (>0.90), TLI (>0.90), RMSEA (<0.08), and SRMR (<0.08). The Bollen–Stine bootstrap *p*-value was also examined, where applicable, as a robust indicator of model fit. Modification indices (M.I.) were cautiously reviewed to guide theoretically justifiable respecifications, such as correlating error terms between items with clear semantic overlap or removing items causing significant localized strain. Finally, the internal consistency of the purified scales (or subscales) was assessed using Composite Reliability (CR) or Cronbach’s alpha (*α* > 0.70).

#### 2.3.2. Measurement Model Estimation and Validation Procedures

Prior to model validation, data preprocessing and preliminary tests were conducted. To assess the risk of common method bias (CMB), procedural methods such as anonymous questionnaires and randomization of item order were implemented during the study design phase. A post hoc Harman’s single-factor test indicated that the first unrotated factor accounted for only 34.051% of the total variance, which is below the commonly recommended 40% threshold. Additionally, an attempt to estimate a model including an unmeasured Latent Method Factor (LMF) failed to converge due to model over-specification, a challenge commonly encountered in complex CFA models ([25]). These results suggest that CMB is not a significant concern in this study. Additionally, collinearity diagnostics were conducted for the moderated mediation models. The results confirmed that all variance inflation factor (VIF) values were well below the 10.0 threshold: for the Model 1 and 2 predictors (based on CA, SEAIone, and AIA), the highest VIF was 6.846; for the Model 3 predictors (based on CA, SEAItwo, and AIA), the highest VIF was 8.273. This indicates that multicollinearity was not a significant concern for hypothesis testing.

After confirming data quality, we proceeded to validate the overall measurement model using Confirmatory Factor Analysis (CFA) in AMOS 26.0, employing Maximum Likelihood (ML) estimation. Given the significant multivariate non-normality identified in the data (Mardia’s c.r. = 27.212), we also utilized bootstrapping (5000 samples) to generate robust bias-corrected confidence intervals (BCa CI) for parameter estimates. Model fit was evaluated using several fit indices: *χ*^2^/*df* (<3), CFI (>0.90), TLI (>0.90), RMSEA (<0.08), and SRMR (<0.08) ([9]). Additionally, the Akaike Information Criterion (AIC) was used to compare competing models.

Specifically, we compared a baseline single-factor model with our proposed theoretical model, which consists of 10 distinct first-order factors structured under 5 main constructs: Career Adaptability (CA; specified as a second-order factor with four first-order factors: Concern, Control, Curiosity, and Confidence), Job Satisfaction (JS; two first-order factors: Extrinsic Satisfaction and Intrinsic Satisfaction), Work Engagement (WE), AI Anxiety (AIA), and AI Self-Efficacy (SEAI; two correlated first-order factors: Comfort with AI and AI Technological Skills). Discriminant validity was confirmed by examining the 95% bias-corrected bootstrap confidence intervals (BCa CI) for the correlations between the latent factors, ensuring that no interval included 1.0. 

To test our complex hypotheses, we employed a two-stage analytical strategy. This approach allowed us to first confirm the foundational latent relationships before testing the nuanced conditional effects.

Stage 1: Latent Variable Mediation (AMOS)

First, we constructed a latent variable mediation model in AMOS 26.0 to confirm the relationships between the measurement-error-free constructs. This model simultaneously tested the base structural paths from the unified Career Adaptability (CA Total Score) latent factor to the two Job Satisfaction factors (JS1: Extrinsic, JS2: Intrinsic) and subsequently to Work Engagement (WE) while controlling for measurement error. This initial stage utilized latent factors and 95% bias-corrected bootstrap confidence intervals (BCa CI) to confirm the overall viability of the total CA score in the foundational mediation pathways.

Stage 2: Moderated Mediation Analysis (PROCESS)

Following the confirmation of the base relationships in AMOS, the core moderated mediation hypotheses were tested using [12]’ ([12]) PROCESS macro (v4.2) for SPSS. Model 11 was specifically employed to analyze the complex conditional indirect effects. The predictor variable (X) in all subsequent models was the CA Total Score. The significance of all conditional indirect effects was tested using the bias-corrected percentile bootstrap method, with 5000 bootstrap samples and a 95% confidence interval ([13]). Multicollinearity was assessed via VIF/Tolerance diagnostics. These models examined three distinct, significant moderated mediation pathways:(1)Pathway 1 (Extrinsic Focus, SEAI1): The indirect effect of CA (Total Score) on Work Engagement through Extrinsic Satisfaction (JS1), jointly moderated by SEAI1 and AIA.(2)Pathway 2 (Second SEAI Dimension): The indirect effect of CA (Total Score) on Work Engagement through Extrinsic Satisfaction (JS1), jointly moderated by SEAI2 and AIA.(3)Pathway 3 (Intrinsic Focus, SEAI1): The indirect effect of CA (Total Score) on Work Engagement through Intrinsic Satisfaction (JS2), jointly moderated by SEAI1 and AIA. Standardized effect sizes (ΔR^2^ for interaction terms) were reported for significant interaction effects, and simple slopes/interactions were graphed. Age, ethnicity, marriage, education, income, and industry were included as control variables in all hypothesis testing models.

## 3. Results

The results evaluated the hypotheses and validated the measurement model. Firstly, we validated the model with confirmatory factor analysis (CFA), comparing the proposed ten-factor specification with a baseline single-factor alternative; the results provided strong evidence of discriminant validity among the constructs. Then, we tested the core hypotheses—including the moderated mediation model—using structural equation modeling (SEM) and [12]’ ([12]) PROCESS macro. The subsections that follow detail each stage of the analysis, including measurement-model validation, descriptive statistics, and correlation analyses.

### 3.1. Measurement Model Validation

Following the procedures outlined in the Section 2.3, the overall measurement model was estimated using AMOS 26.0.v. This model incorporated a 10 first-order factor structure, structured under 5 main constructs: Career Adaptability (CA; specified as a second-order factor with four first-order factors: Concern, Control, Curiosity, and Confidence), Job Satisfaction (JS; two first-order factors: Extrinsic Satisfaction and Intrinsic Satisfaction), Work Engagement (WE; one first-order factor), AI Self-Efficacy (SEAI; two correlated first-order factors: Comfort with AI and AI Technological Skills), and AI Anxiety (AIA; one first-order factor), with all latent factors allowed to covary.

As shown in Table 2, the proposed 10-factor measurement model demonstrated an excellent fit to the data (*χ*^2^(1916) = 2874.785, *χ*^2^/*df* = 1.500, CFI = 0.928, TLI = 0.924, RMSEA = 0.040 [90% CI: 0.037, 0.043], SRMR = 0.0605), meeting the established criteria ([9]; [40]). This model showed a significantly better fit than the baseline single-factor model, which exhibited poor fit (*χ*^2^(1941) = 6055.792, *χ*^2^/*df* = 3.120, CFI = 0.690, TLI = 0.678, RMSEA = 0.083 [90% CI: 0.080, 0.085]), SRMR = 0.0961. The superiority of the proposed model was confirmed by a significant chi-square difference test (*χ*^2^(25) = 3181.007, *p* < 0.001) and substantially lower information criteria (AIC = 3202.785 vs. 6333.792; BIC = 3816.111 vs. 6853.623).

Finally, discriminant validity was supported. Examination of the 95% bias-corrected bootstrap confidence intervals for the correlations between the ten latent constructs confirmed that none of the intervals included 1.0, indicating that the constructs are empirically distinct. Despite some substantial correlations (e.g., Work Engagement and Intrinsic Satisfaction (JS2), *r* = 0.848, 95% CI [0.796, 0.889]; Extrinsic (JS1) and Intrinsic Satisfaction (JS2), *r* = 0.809, 95% CI [0.758, 0.852]), these constructs were empirically distinguishable in our data.

These results provide strong evidence for the construct and discriminant validity of the overall measurement model, establishing a robust foundation for subsequent hypothesis testing.

### 3.2. Descriptive Statistics and Correlation Analysis

The results of the descriptive statistics are presented in Table 3.

The moderated moderation hypothesis was tested using Model 11 of the PROCESS macro for SPSS (v4.0; [12]). Considering the sample size of this study and balancing the need to address extreme outliers with preserving statistical power, we adopted a relatively strict criterion, identifying and removing cases with absolute Z-scores greater than 2.96 on key variables. Based on this criterion, Extreme Values (70) were removed from the original sample (9880). The proportion of removed extreme values is 0.728%.

A total of 5000 bootstrap samples were used to estimate the 95% bias-corrected confidence intervals (BCa CI). In the models, Career Adaptability (CA) was the independent variable (X), Job Satisfaction dimensions (JS1 or JS2) were the mediator (M), AI Self-Efficacy dimensions (SEAI1 or SEAI2) were the primary moderator (W), AI Anxiety (AIA) was the second-stage moderator (Z), and Work Engagement (WE) was the dependent variable (Y). Covariates included in the models were education level, industry, and occupation level. The conditional indirect effects were subsequently examined at three levels of the two moderators (mean ± 1 SD). During the analysis, we systematically examined the operationalization of the mediator, Job Satisfaction (JS), by testing the hypotheses separately for the Extrinsic Satisfaction (JS1) and Intrinsic Satisfaction (JS2) dimensions, resulting in three distinct moderated mediation pathways. The results showed that the hypothesized core mechanism—the complex three-way interaction effect (X × W × Z on the mediation path)—was statistically significant for multiple pathways, providing robust support for our conditional model.

Specifically:

Model 1 (CA → JS1 ← W: SEAI1): For the CA → JS1 → WE pathway, with Comfort with AI (SEAI1) specified as the primary moderator (W), the key three-way interaction effect (CA × SEAI1 × AIA) predicting the mediator, Extrinsic Satisfaction (JS1), was statistically significant (*b* = −0.3509, *p* = 0.0075).

Model 2 (CA → JS1 ← W: SEAI2): For the CA → JS1 → WE pathway, with AI Technological Skills (SEAl2) as the primary moderator (W), the three-way interaction effect (CA × SEAl2 × AIA) predicting JS1 was also significant (*b* = −0.3502, *p* = 0.0172).

Model 3 (CA → JS2 ← W: SEAI1): For the CA → JS2 → WE pathway, with Comfort with AI (SEAI1) as the primary moderator (W), the three-way interaction effect predicting JS2 was also significant (*b* = −0.2171, *p* = 0.0428).

Given the significant three-way interactions found across all three theoretically relevant pathways, all three models were retained for primary analysis to provide a comprehensive understanding of the mechanisms. For the primary moderator, AI Self-Efficacy (SEAI), we systematically tested both the “Comfort with AI” (SEAI1) and “AI Technological Skills” (SEAI2) dimensions, corresponding to the pathways identified above. The results indicated that the moderating effect was statistically significant for both factors, depending on the specific path.

The core three-way interaction effect was significant when “Comfort with AI” (SEAI1) was used as the moderator (W) for both the CA → JS1 path (*b* = −0.3509, *p* = 0.0075) and the CA → JS2 path (*b* = −0.2171, *p* = 0.0428). Similarly, the three-way interaction was significant when “AI Technological Skills” (SEAI2) was used as the moderator (W) for the CA1 → JS2 path (*b* = −0.3500, *p* = 0.0172). Therefore, our main analysis reports on all three of these significant models to precisely test the hypothesized mechanisms.

This systematic comparison yielded three significant moderated mediation models, which are detailed below:

Figure 1a shows Model 1: X = Carrer Adaptability (CA), M = Extrinsic Satisfaction (JS1), W = Comfort with AI (SEAI1) (*n* = 311).

Figure 1b shows Model 2: X = Carrer Adaptability (CA), M = Extrinsic Satisfaction (JS1), W = AI Technological Skills (SEAI2) (*n* = 311).

Figure 1c shows Model 3: X = Carrer Adaptability (CA), M = Intrinsic Satisfaction (JS2), W = Comfort with AI (SEAI1) (*n* = 311).

The conditional indirect effect was examined at three levels of the moderators (mean ± 1 SD). We conducted three separate tests of our moderated moderation model, corresponding to the three significant pathways identified in our analysis.

Hypotheses 1, 2, 7, and 8 (H1, H2, H7, H8): Mediation and Direct Effect Testing.

These hypotheses predicted the main direct and mediating effects. We examined these paths within the full models. H8 (JS → WE) was consistently supported. In the second stage of the models, both Intrinsic Satisfaction (JS2) (e.g., *b* = 0.8684, *p* < 0.0001) and Extrinsic Satisfaction (JS1) (*b* = 0.5400, *p* < 0.0001) significantly and positively predicted Work Engagement.

H7 (CA → WE) was also consistently supported. The direct effect of Career Adaptability (CA) on Work Engagement remained significant in all three models (e.g., *b* = 1.2912, *p* < 0.0001; *b* = 0.7392, *p* < 0.0001).

H1 and H2 (CA → JS) were not supported as simple positive main effects. The main effect of CA on the mediators, when moderators were at zero, was either non-significant (CA → JS2: *b* = −2.9548, *p* = 0.1014) or significant and negative (e.g., CA → JS1: *b* = −4.9042, *p* = 0.0265). This pattern provides strong support for the foundational mediation pathways, as their strengths are qualified by the higher-order interactions explored next.

Hypotheses 3, 5, and 11 (H3, H5, H11): Two-Way Moderation Effect Testing.

These hypotheses predicted the two-way interactions influencing the mediators (JS1 or JS2).

H3 (CA × SEAI) was consistently supported across all three models. The CA × SEAI1 (Int_1) interaction significantly predicted JS1 (*b* = 1.3880, *p* = 0.0014). The CA × SEAI2 (Int_1) interaction significantly predicted JS1 (*b* = 1.5080, *p* = 0.0006). The CA × SEAI1 (Int_1) interaction significantly predicted JS2 (*b* = 1.0223, *p* = 0.0039).

H5 (CA × AIA) was supported in two of the three models. The interaction significantly predicted JS1 in both Model 1 (*b* = 1.4051, *p* = 0.0298) and Model 2 (*b* = 1.0282, *p* = 0.0313), though it was not significant in the model predicting JS2 (*b* = 0.7756, *p* = 0.1417).

H11 (SEAI × AIA) was supported in all three models (Model 1: *b* = 1.3401, *p* = 0.0093; Model 2: *b* = 1.3196, *p* = 0.0208; Model 3: *b* = 0.9571, *p* = 0.0230). These results must be interpreted in light of the significant three-way interactions reported below.

Hypotheses 4 and 6 (H4, H6): Moderated Moderation Effect Testing.

Hypotheses H4 and H6 predicted that the CA → JS → WE indirect effect is conditional upon both SEAI and AIA. Consistent with these hypotheses, and as shown in Table 4a, the key three-way interaction (Int_4: X × W × Z) predicting the mediator was significant across all three models.

Model 1 (CA → JS1 ← W: SEAI1): The three-way interaction predicting JS2 was significant (*b* = −0.3509, *p* = 0.0075), explaining a significant increment in the mediator’s variance (ΔR^2^ = 0.0148, F(1, 300) = 7.2478, *p* = 0.0075). Most importantly, the Index of Moderated Moderated Mediation was significant (Index = −0.1895, 95% BCa CI [−0.4243, −0.0584]).

Model 2 (CA → JS1 ← W: SEAI2): The three-way interaction predicting JS1 was significant (*b* = −0.3502, *p* = 0.0172), explaining a significant increment in the mediator’s variance (ΔR^2^ = 0.0115, F(1, 300) = 5.7398, *p* = 0.0172). The overall Index of Moderated Moderated Mediation was also significant (Index = −0.1891, 95% BCa CI [−0.3901, −0.0575]).

Model 3 (CA → JS2 ← W: SEAI1): The three-way interaction predicting JS2 was significant (*b* = −0.2171, *p* = 0.0428), explaining a significant increment in the mediator’s variance (ΔR^2^ = 0.0073, F(1, 300) = 4.1375, *p* = 0.0428). The overall Index of Moderated Moderated Mediation was significant (Index = −0.1886, 95% BCa CI [−0.5106, −0.0156]). To visualize these significant three-way interactions, simple slopes were plotted for the effect of Career Adaptability on Job Satisfaction at three levels of AI Anxiety (see Figure 2).

To fully assess this effect, we examined the relevant mediation indices (see Table 4b).

For (CA → JS1 ← W: SEAI1), further analysis of the Indices of Conditional Moderated Mediation (testing the Career Adaptability (CA) × Comfort with AI (SEAI1) interaction at different levels of AI Anxiety (AIA)) revealed that the moderating effect of Comfort with AI (W) on the indirect path was positive and significant at low levels of AI anxiety (Z) (Index = 0.6083, 95% BCa CI = [0.2407, 1.1574]), and this effect remained significant at moderate (Index = 0.4654, 95% BCa CI = [0.1737, 0.8775]) and high levels of AI anxiety (Index = 0.3225, 95% BCa CI = [0.0602, 0.6212].

For Model 2 (CA → JS1 ← W: SEAI2), the “Indices of conditional moderated mediation” revealed a similar pattern: the moderating effect of AI Technological Skills (SEAI2) (W) was significant at low (Index = 0.6734, 95% BCa CI = [0.3097, 1.0930]) and moderate (Index = 0.5307, 95% BCa CI = [0.2263, 0.8559]) levels of AI anxiety, and remained significant at high levels of AI anxiety (Index = 0.3881, 95% BCa CI = [0.1050, 0.6476]). The conditional indirect effects are detailed in Table 5b.

For Model 3 (CA → JS2 ← W: SEAI1), the conditional indices (Table 4b) showed that the moderating effect of Comfort with AI (SEAI1) (W) was significant at low (Index = 0.7472, 95% BCa CI = [0.2825, 1.5341]) and moderate (Index = 0.6050, 95% BCa CI = [0.2342, 1.1938]) levels of AI anxiety, and was also significant at high levels (Index = 0.4628, 95% BCa CI = [0.1361, 0.8738]). The conditional indirect effects are detailed in Table 5c.

This pattern was further validated in the final analysis of the conditional indirect effects. We focus here on Model 3 (CA → JS2 ← W: SEAI1), as this pathway through Intrinsic Satisfaction (JS2) was confirmed as the most robust mediating mechanism in subsequent SEM analysis (see Table 5c). The results showed that the indirect effect of Career Adaptability (CA) on Work Engagement (mediated by JS2) was significant and positive across all nine conditions examined44. The strength of this indirect effect was contingent upon the interaction between Comfort with AI (SEAI1) (W) and AI Anxiety (AIA) (Z), consistent with the moderated moderation hypothesis.

Specifically, under low AI Anxiety (Z = Low), the positive indirect effect strengthened significantly as Comfort with AI (W) increased (Effect increased from 1.1310 at Low W to 2.1365 at High W; the contrast between these effects was significant, Contrast = 1.0055, 95% BCa CI [0.3802, 2.0644]). In contrast, under high AI Anxiety (Z = High), this enhancing effect of Comfort with AI was weaker, but remained significant (Effect = 0.9309 at Low W vs. 1.5537 at High W; the contrast between these effects was also significant, Contrast = 0.6228, 95% BCa CI [0.1832, 1.1758]). This confirms that while high AI anxiety dampens the moderating effect of AI self-efficacy, it does not nullify it.

### 3.3. Latent Variable Structural Equation Modeling (SEM) Analysis for Mediation Paths

Following the preliminary exploration using PROCESS, we conducted a latent variable structural equation modeling (SEM) analysis using AMOS (Version 26.0). This served as a crucial step to provide more accurate, disattenuated estimates for the core mediation paths (H1, H2) proposed in our model. By explicitly modeling latent constructs, SEM offers a more robust validation of the mediating roles of Extrinsic and Intrinsic Job Satisfaction identified as potentially significant in the initial PROCESS analyses.

#### 3.3.1. Model Specification and Estimation

We specified a latent multiple mediation model (see Figure 3) where the second-order latent factor of Career Adaptability (CA) predicted latent Work Engagement (WE) both directly and indirectly through the two latent mediators: Extrinsic Satisfaction (JS1) and Intrinsic Satisfaction (JS2). The latent CA factor was defined by its four first-order factors (Concern, Control, Curiosity, and Confidence). Latent variables were defined by their respective indicators, consistent with prior measurement model validation.

Given the multivariate non-normality noted in the data (Mardia’s coefficient c.r. = 25.909), Maximum Likelihood (ML) estimation was employed, coupled with bias-corrected bootstrapping (based on 5000 resamples). This approach generated robust standard errors and 95% confidence intervals (CI) for parameter estimates, particularly the indirect effects, thereby ensuring that inferences were not unduly compromised by violations of normality assumptions.

#### 3.3.2. Model Fit

The adequacy of the specified latent structural model was assessed using standard fit indices. The results suggested an acceptable fit to the data: the chi-square/degrees of freedom ratio was within acceptable limits (*χ*^2^/*df* = 1815.437/1157 = 1.569); both the Comparative Fit Index (CFI = 0.939) and the Tucker–Lewis Index (TLI = 0.936) surpassed the conventional threshold of 0.90; and the Root Mean Square Error of Approximation (RMSEA = 0.043, 90% CI [0.039, 0.047], PCLOSE = 0.999) indicated a good fit according to criteria suggested by [15] ([15]). Although the model’s chi-square was significant (*p* = 0.000), which is common in large samples, the convergence of the other key indices provided sufficient support for interpreting the model’s path coefficients.

#### 3.3.3. Hypothesis Testing: Mediation Effects in the Latent Model

Table 6 displays the unstandardized path coefficients (*b*) and significance tests based on the bootstrap results. The SEM analysis, controlling for measurement error, provided a more refined picture of the mediation processes.

The path analysis (Table 6) revealed that the second-order latent Career Adaptability (CA) factor strongly and positively predicted both latent mediators: Extrinsic Satisfaction (JS1; *b* = 1.249, BCa 95% CI [0.901, 1.885]) and Intrinsic Satisfaction (JS2; *b* = 1.361, BCa 95% CI [1.023, 1.967]).

A critical distinction emerged in the paths linking the mediators to latent Work Engagement (WE). While latent Intrinsic Satisfaction (JS2) significantly and positively predicted WE (*b* = 0.716, BCa 95% CI [0.548, 0.912]), the path from latent Extrinsic Satisfaction (JS1) to WE was non-significant (*b* = 0.107, BCa 95% CI [−0.030, 0.250]). This finding clarifies the mediation mechanism, suggesting that the effect of adaptability on engagement flows primarily through intrinsic, rather than extrinsic, satisfaction. This discrepancy suggests that the preliminary PROCESS findings related to Extrinsic Satisfaction may have been influenced by measurement error.

Regarding the direct effect (path c′), the path from latent Career Adaptability (CA) to WE remained significant (*b* = 0.715, BCa 95% CI [0.385, 1.396]), even after accounting for the mediators.

Consistent with these path coefficients, the bootstrap analysis of the total indirect effect (Table 7) confirmed the overall mediation mechanism. The total indirect effect (linking latent CA to WE via both mediators) was statistically significant (Total *b* = 1.109, BCa 95% CI [0.796, 1.598]). Given the non-significant JS1 → WE path (*b* = 0.107, BCa 95% CI [−0.030, 0.250]), this robust indirect effect is confirmed to be channeled exclusively through Intrinsic Satisfaction (JS2), which had a significant path to WE (*b* = 0.716, BCa 95% CI [0.548, 0.912]).

In conclusion, the SEM analysis provides crucial validation and refinement of the mediation hypotheses (H1/H2). By accounting for measurement error, the results offer robust support for Intrinsic Satisfaction (JS2) as the significant and primary mediator linking Career Adaptability (CA) to Work Engagement. Contrary to implications from preliminary analyses using composite scores, the mediating role of Extrinsic Satisfaction (JS1) was not supported in the latent variable framework. This underscores that the beneficial impact of career adaptability on engagement predominantly operates by enhancing employees’ intrinsic satisfaction with their work. This clarified mediation pathway, along with the significant direct effect of Career Adaptability (CA) on Engagement (*b* = 0.715, BCa 95% CI [0.385, 1.396]), provides a robust, disattenuated foundation for subsequent discussion and interpretation.

## 4. Discussion

With the increasingly accelerated pace of artificial intelligence (AI) entering the workplace, how employees manage to adopt and maintain positive work-related consequences has become a vital issue. This study attends to digital natives in early career stages and intends to identify the underlying processes that explain how CA is related to WE. More precisely, we investigated that job satisfaction (JS) mediated and the intricate moderation of AI self-efficacy (SEAI) as well as AI anxiety (AIA). The integration of early tests with manifest variables (as implemented in the PROCESS macro) and more sophisticated latent variable SEM, provided a number of interesting results, but also identified several sources of complexity and methodological considerations. Taken together these findings have implications for understanding individual adaptation in the AI-enabled workplace at deeper level and it contributes to both theoretical lenses—i.e., Conservation of Resources (COR) theory and Social Cognitive Theory (SCT)—as well as practical implications for organizational practice.

The Centrality of Intrinsic Satisfaction (JS2) as the Key Mediator

Based on the SEM results, this study reveals distinct effects of career adaptability (CA) on work engagement in an AI-driven environment. A key finding from our latent variable model (see Table 6) is that the influence of CA on WE flows through two distinct paths: a significant direct effect (*b* = 0.715, *p* < 0.001), and a significant indirect effect that is channeled exclusively through Intrinsic Job Satisfaction (JS2) (*b* = 0.716, *p* < 0.001). It suggests that intrinsic job satisfaction (JS2) highlights the critical role of intrinsic satisfaction in driving employee engagement. This aligns with prior research that has shown a positive association between CA and work outcomes (e.g., [27]; [4]). This aligns with H1, H2, H7, and H8. Crucially, the mediating path through Extrinsic Job Satisfaction (JS1) was found to be non-significant (JS1 to WE: *b* = 0.107, *p* = 0.116). This suggests that in the AI era, the ability to build meaningful interpersonal relationships, gain autonomy, and utilize skills at work are key factors that enhance intrinsic satisfaction, which, in turn, fosters work engagement. This finding challenges prior research that often uses global satisfaction variables, which may mask the distinct contributions of different satisfaction dimensions.

In contrast, while flexibility may improve extrinsic satisfaction (e.g., JS1), this dimension does not significantly influence work engagement when controlling for measurement error in our SEM model. This underscores the central role of intrinsic rewards—such as work meaning, autonomy, and skill use—in motivating employee commitment and performance. Our findings challenge prior research that often uses global satisfaction variables (e.g., [16]), which may mask the distinct contributions of different satisfaction dimensions. Here, we demonstrate that career adaptability specifically enhances intrinsic satisfaction, which then translates into higher work engagement. While extrinsic rewards (e.g., pay, promotion) may offer short-term satisfaction, it is intrinsic rewards that sustain long-term engagement and performance.

This result is consistent with Conservation of Resources (COR) theory ([14]). Our research demonstrates that Career Adaptability, validated as a unified, second-order resource, invests in psychological resources (JS) to generate gains (WE). However, the “return on investment” is overwhelmingly channeled through intrinsic psychological gains (JS2). The boost in extrinsic satisfaction (JS1) provided by CA did not significantly translate into work engagement in the robust latent model. This implies that in a rapidly evolving technological landscape, employees’ sustained engagement may depend less on traditional extrinsic motivators and more on the intrinsic value and fulfillment their work provides. These findings offer a psychological mechanism that helps explain why adaptable individuals are better equipped to “thrive” in the face of technological change, rather than merely “survive”. This also aligns with [24]’s ([24]) call for further exploration of the mechanisms underlying career adaptability in AI-driven environments.

When ‘Technical Confidence’ Meets ‘Existential Angst’: AI Anxiety Dampens, but Does Not Nullify, the Benefits of AI Self-Efficacy

This study takes a thought-provoking and cautious perspective on the cognitive-affective dynamics of artificial intelligence (AI) adoption, with a particular focus on the moderating roles of AI self-efficacy (SEAI) and AI anxiety (AIA) (Hypotheses H3–H6, H11). The preliminary analysis, conducted using the PROCESS macro on observed variables, revealed significant three-way interaction effects in specific models. Our findings show a consistent pattern: the positive moderating effect of AI Self-Efficacy (W) on the indirect pathway from career adaptability (CA) to job satisfaction (JS) is contingent upon the level of AI Anxiety (Z). Specifically, the enhancing effect of SEAI was strongest at low levels of AIA and progressively weaker at moderate and high levels of AIA. This revises a common “nullification” hypothesis; our results indicate that high anxiety dampens or suppresses the benefits of high self-efficacy, but it does not completely eliminate them. The positive moderating effect of SEAI, though diminished, remained statistically significant even under high anxiety. This observed phenomenon is consistent with Social Cognitive Theory (SCT; [3]), which asserts that intense negative affective states like anxiety can hinder the demonstration of efficacy beliefs.

Beyond this, the pattern of findings here further correspond to a body of AI research that emphasizes the benefits associated with SEAI ([21]) and costs incurred by AIA ([35]; [39]). From a humanization view, this finding illuminates an important psychological challenge confronting employees in the age of AI; that is, even if people acquire technical ability and develop self-belief (high SEAI), other more fundamental insecurities(for example, job losing/machine alienation feelings, or skill decay) might dampen these positive effects of self-efficacy. It could prevent the appropriate translation of the adaptability resources into job satisfaction. Thus, in the era of AI, psychological safety (a subject’s confidence about “not getting things wrong”) may be a precondition for technical competency (“proficiency of skill”) such that its benefits can be fully realized.

However, it is important to emphasize with scientific rigor that there are key differences between the findings in this process and our more robust SEM latent variable analysis results. While the process shows complex interactions, our SEM analysis (Table 6) shows that some of the underlying paths upon which these models rely are invalid after controlling for measurement error. The most striking example is that the JS1 → WE path is insignificant, casting doubt on the validity of the entire moderating mediation chain. This underscores the need for future research, using more advanced methods (such as LMS), to rigorously investigate, while accounting for measurement error, whether such complex interactions between AI self-efficacy and AI anxiety truly exist and how they influence employee adaptation and well-being. Such efforts are essential not only for advancing theories, such as Social Cognitive Theory in the AI context, but also for informing effective organizational support strategies.

Measurement Reflections—Validating Complex Constructs

The strict measurement validation (MV) process employed in this study provided significant insights into the measurement of constructs.

First, a significant finding was the successful validation of Career Adaptability as a second-order construct. While the initial 24-item scale required refinement (resulting in 18 items), the final second-order model fit the data excellently. This provides strong support for treating CA as a unified, higher-order psychological resource in this context, justifying its use as a single latent factor in our SEM and as a total score in PROCESS.

Second, the strong association between Job Satisfaction (JS1) and its facets, along with the functional separation of these satisfaction facets through Structural Equation Modeling (SEM), underscores the theoretical importance of distinguishing between different forms of satisfaction in research ([18]). This finding also highlights potential challenges arising from measurement overlap, which may explain the difficulty in defining overall job satisfaction as a well-defined latent construct.

Third, the significant correlations observed between the dimensions of Work Engagement supported the argument for treating it as a unidimensional construct in this context. Since this study conducted in the AI context, this finding suggests that the abovementioned variables may instead function as a situational behavioral tactic. This result may inform the future development of measurement scales under AI context.

### 4.1. Theoretical Contributions: Extending Theories of Adaptation, Motivation and Cognitive-Affective Interaction in the AI Domain

Given that we have summarized the findings and reflections in this paper, our current study provides several theoretical contributions, especially for applying established theories to the novel work setting AI at workplace:(1)Expanding COR Theory Application in AI Transformation by Emphasizing the Centrality of “Intrinsic Returns”: Adding to the extant literature that has only established adaptability as a resource ([27]), our empirical study extends it through SEM, demonstrating that intrinsic job satisfaction (JS2) is the core mechanism for transforming career adaptability (CA) into work engagement (WE). This study offers a focused psychologically based empirical model for the COR (Conservation of Resources) theory’s cycle of investment and gain in resources ([14]) in technology—highlighting how intrinsic work quality is pivotal to forming spirals of conservation and gain. It therefore operates as a qualifier, or modifier, to views that overstate the importance of extrinsic motivation.(2)Implications and Directions for Future SCT-Based Research on Cognitive-Affective Complexity in AI Adaptation: Although the proposed interaction effects were not upheld, the early pattern of results from PROCESS add to our understanding within Social Cognitive Theory (SCT) regarding person–behavior–environment interactions (especially in the AI context) ([3]; [33]). This result warns against univariate additive models, where the positive influence of SEAI ([21]) and the negative impact of AIA ([35]) are assumed to directly add. It instead suggests the possibility of multiple non-linear relationships between cognitive beliefs (SEAI) and affective responses (AIA) for AI adoption. The preliminary findings and methodological reflections presented here strongly encourage future work using more refined (e.g., Latent Variable Modeling) methods to better understand the mechanisms of SCT in the new AI adaptation domain.(3)Reinforcing the Importance of Measurement Precision and Methodological Competency in AI Research. Given the discrepancy between the PROCESS and SEM results, as well as contrasting mediating roles of extrinsic job satisfaction (vs. intrinsic job satisfaction), the empirical investigation proves that construct dimensionality should be dealt with, allowing low-level measurement error to be reduced when researching the consequences of AI effects. Composite scores alone mask important mechanisms and may even result in false conclusions based on statistical artifacts. This study serves as a reminder that AI research in relation to employee psychology should pursue more elaborate theoretical models (per the multistage model, at least) and careful measurement choices, methodological reflexivity (e.g., what methods are suited for PROCESS exploration vs. SEM confirmation), and an unwavering aim for improved construct validity.

### 4.2. Practical Relevance: From Training Skills to Human-Centered AI Management

Building on the significant mediation effects of intrinsic satisfaction and the tentative speculations regarding the interaction between anxiety and confidence, we offer several strategic, human-centered recommendations for managers in organizations, particularly when guiding young employees through AI transitions:

Elevate “Job Quality Enhancement” as a Foundational Strategy for AI-Era Talent Management: Given that intrinsic satisfaction is the key lever for unlocking the engagement dividend of adaptability, organizations planning to adopt AI should prioritize the redesign of jobs to offer intrinsically rewarding experiences. This approach moves beyond a focus on simple efficiency gains, addressing questions such as: How can AI enhance, rather than diminish, employees’ sense of autonomy, skill variety, and task significance? How can human–machine collaboration be designed to be engaging and creative? In what ways can AI facilitate employee learning and development, helping individuals find greater fulfillment in their professional roles? Additionally, how can technology support team unity and social interaction instead of hindering them? Shifting the focus from technical training to the qualitative aspects of work may prove to be a more fundamental strategy for fostering engagement among younger employees.

Operationalize a Balanced “Empowerment and Reassurance” Approach to Facilitating AI Adaptation: Given the complex, tentative interaction effects observed, and the potential risk of high anxiety undermining the benefits of excessive confidence, organizations must not only focus on skill development but also ensure employees’ psychological safety during the AI adoption process. This includes providing high-quality, context-relevant AI training (building SEAI), fostering a positive climate for AI use, and addressing psychological safety needs. Practical steps might involve: continuous, transparent communication about AI strategies, impacts, and boundaries; clear AI ethics guidelines and protection policies for employees; offering psychological counseling and stress management resources; promoting empathic leadership that actively listens to employee concerns; and implementing performance evaluation systems that emphasize AI as a tool to augment human capabilities rather than replacing humans with AI. We argue that fostering a sense of positive future expectations and psychological safety around AI should be key performance indicators (KPIs) for successful AI integration.

Cultivate Career Adaptability as a Holistic Meta-Competency: This research reaffirms the importance of career adaptability as a meta-competency strongly linked to positive work outcomes ([28]; [23]; [27]; [22]). However, it is important to note that the impact of these recommendations may vary based on individual roles, industries, or the extent of AI integration within a given organization. Managers should tailor these strategies to their specific context. Encouraging cross-functional experiences, fostering reflection, establishing effective mentoring and peer support mechanisms, and cultivating a growth-oriented mindset will prepare employees to navigate uncertainty, learn deliberately, respond effectively, and derive meaning as they transition through career changes driven by AI, rather than being passive recipients of these changes.

However, it is important to note that the impact of these recommendations may vary based on individual roles, industries, or the extent of AI integration within a given organization. Managers should tailor these strategies to their specific context.

## 5. Limitations and Future Research Directions

A comprehensive understanding of the limitations of this study is essential for accurate scientific interpretation and for guiding future research directions.

(1)Innate Limitation for Causal Inference—Cross-Sectional Design: This is the most significant limitation, as the cross-sectional design prevents us from confirming the directionality or causality implied by our mediation model. Future research should prioritize longitudinal designs (e.g., multi-wave panel studies) to track these relationships over time and test the proposed causal chain. Additionally, experimental or quasi-experimental designs (e.g., comparing employee transitions before and after AI implementation in organizations with different support structures) would provide stronger causal evidence.(2)Sample and Contextual Limitations: The sample in this study is limited to young Chinese employees (aged 18–25), which restricts the generalizability of the findings to a broader population, such as older workers or individuals from diverse cultural backgrounds. Furthermore, the impact of AI is not uniformly distributed across organizations. Replication of these findings in more diverse samples and contexts is necessary, with explicit testing of the moderating roles of important contextual variables (e.g., AI ethical, leadership style, job autonomy).(3)Common Method Bias (CMB): The reliance on self-report data can inflate relationships. Although procedural checks and Harman’s test were employed and did not indicate a clear dominance of a single factor, this test has limited diagnostic value ([25]). Future research should incorporate multi-source data (e.g., supervisor ratings) or multiple methods (e.g., behavioral indicators, system-logged data) to better address CMB concerns. Additionally, time-lagged designs should be considered to mitigate these biases.(4)The Validation of Complex Constructs and Model Specification: Validating complex psychological constructs, especially adapted or foreign scales, presents inherent challenges. The lack of sufficient participants also limited our ability to perform split-half cross-validation for all scales ([23]). Moreover, the failure to fit an appropriate latent model for Overall Job Satisfaction (JS_All) in AMOS experimentation impeded our ability to test more complex models, particularly those involving JS enrollment in SEM. This issue highlights the importance of distinguishing between JS facets and suggests structural concerns within the JS_All construct. Although the overall SEM fit was acceptable, the significant Bollen–Stine bootstrap *p*-value indicates potential model misfit, which should be considered when interpreting the coefficients.(5)The “Differences in Interactions”—Inconsistent Evidence from PROCESS and SEM: This limitation warrants serious attention. PROCESS analysis consistently found significant three-way interactions. However, more rigorous SEM analysis showed that several key foundational pathways upon which these interactions depend were not significant. This discrepancy suggests that the findings in PROCESS may have been exaggerated by measurement error. This emphasizes the need for future research to employ Latent Moderated Structural Equations (LMS), which is considered the gold standard for testing moderation—especially when latent interactions are included—to obtain more reliable results.(6)The “Tip of the Iceberg”—Model Parsimony and Omitted Variables: Our model focused on a specific set of variables, leaving out potentially important factors. Future studies should aim to develop more comprehensive theoretical models that include additional individual-level variables (e.g., learning agility, personality), task-level factors (e.g., AI complexity, human-AI interaction quality), or organizational influences (e.g., transformational leadership, organizational justice, training effectiveness).

Implications: Future research should address these limitations by conducting longitudinal studies and applying LMS analysis within diverse samples and real organizational settings. This approach will help to fully elucidate the core mechanisms and boundary conditions of employee adaptation in the AI era. Additionally, the exploration of non-linear relationships (e.g., optimal levels of anxiety or efficacy) warrants further investigation.

## 6. Conclusions

This study provides nuanced insights into the relationship between career adaptability and work engagement among young workers in the AI era. Through latent variable modeling, we present strong evidence supporting the proposition that the effect of Career Adaptability (as a unified, second-order construct) on work engagement operates via two distinct mechanisms: a robust direct pathway (CA to WE) and an indirect pathway channeled exclusively through intrinsic job satisfaction (JS2). This underscores the importance of focusing on intrinsic job quality and career adaptability in the context of technological change.

The PROCESS analyses reveal a complex interaction between AI self-efficacy and anxiety, suggesting that high levels of anxiety dampens (but does not nullify) the benefits of high self-efficacy. However, these interaction findings must be interpreted with caution, as our SEM results suggest the mediation pathway through extrinsic satisfaction (JS1) is not robust. As such, it should be verified through more rigorous methods, such as Latent Moderated Structural Equations (LMS), in future studies.

In sum, the results advocate for a more human-centered approach to management in the AI era. It is not enough to foster employee adaptability and technical confidence; attention also be paid to their intrinsic work experiences and psychological safety, with strategies that both “empower” and “reassure.” Future research using advanced longitudinal and latent variable models is necessary to fully explore the interplay between the various dynamics only partially revealed in this study.

## Figures and Tables

**Figure 1 behavsci-15-01682-f001:**
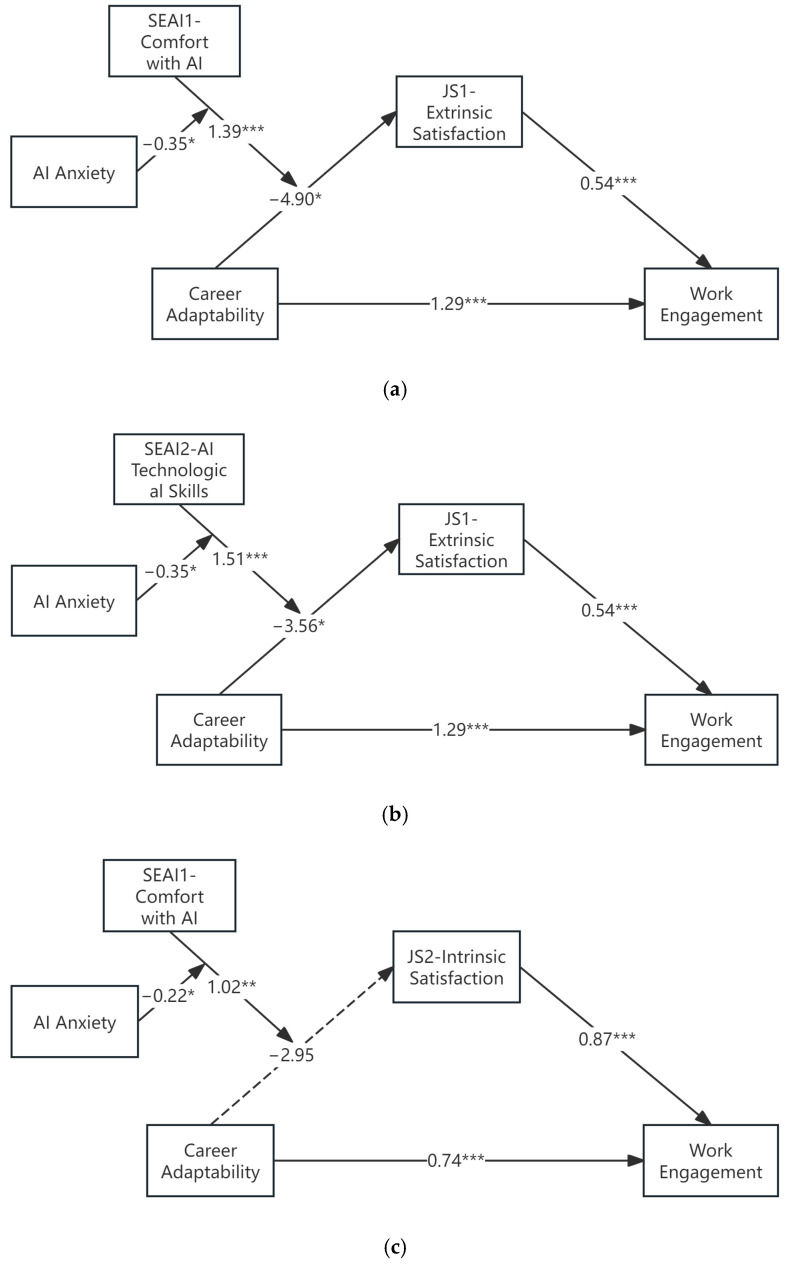
(**a**) Standardized Path Coefficients for the Hypothesized Model. (**b**) Standardized Path Coefficients for the Hypothesized Model. (**c**) Standardized Path Coefficients for the Hypothesized Model. Note: * *p* < 0.05; ** *p* < 0.01; *** *p* < 0.001.

**Figure 2 behavsci-15-01682-f002:**
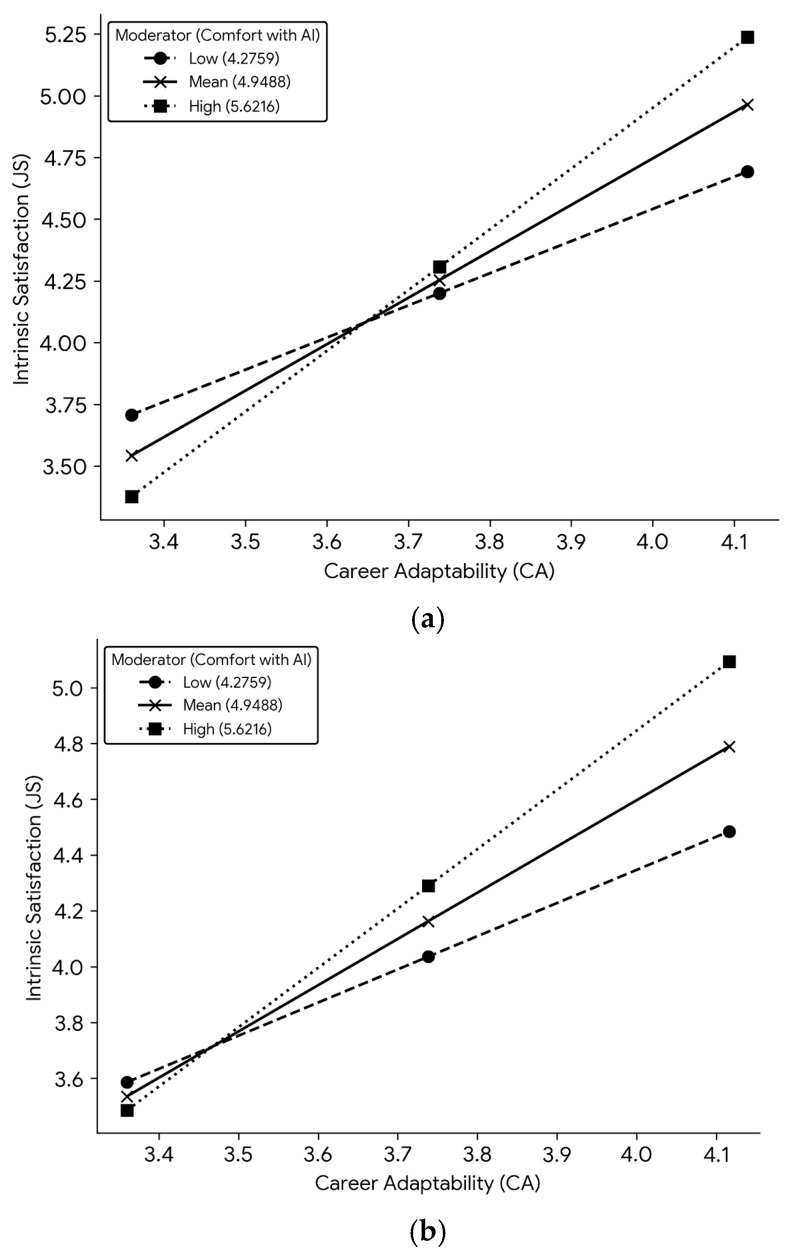
(**a**) Simple Slopes of CA Predicting JS on the low Level of AIA. (**b**) Simple Slopes of CA Predicting JS on the medium level of AIA. (**c**) Simple Slopes of CA Predicting JS on the high level of AIA.

**Figure 3 behavsci-15-01682-f003:**
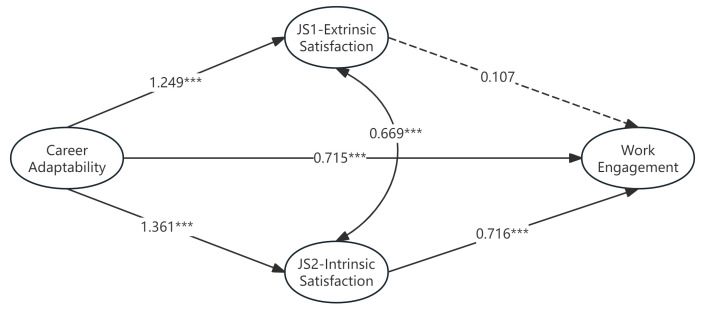
Results of the latent multiple mediation model. Note: Unstandardized path coefficients (*b*) are shown. For clarity, Career Adaptability (CA) is depicted as a single latent construct, though it was specified as a second-order factor in the model. Indicators, error terms, and covariates are omitted. The curved, double-headed arrow represents the covariance between the mediator error terms. * *p* < 0.05, ** *p* < 0.01, *** *p* < 0.001 (based on BCa 95% Confidence Intervals).

**Table 1 behavsci-15-01682-t001:** Reliability Analysis of the Scales.

Construct	Dimension	Items	Cronbach’s α	McDonald’s ω (95% CI)
Career Adaptability	Overall	18	0.857	0.858 [0.835, 0.881]
	Concern	5	0.709	0.717 [0.646, 0.789]
	Control	3	0.769	0.774 [0.734, 0.814]
	Curiosity	6	0.705	0.705 [0.653, 0.757]
	Confidence	4	0.697	0.693 [0.640, 0.766]
Job Satisfaction	Overall	19	0.967	0.968 [0.963, 0.974]
	Extrinsic Satisfaction	14	0.966	0.964 [0.958, 0.970]
	Intrinsic Satisfaction	5	0.894	0.896 [0.878, 0.914]
AI Anxiety	Overall	5	0.850	0.841 [0.812, 0.869]
Self-Efficacy in AI Use (SEAI)	Overall	9	0.855	0.893 [0.875, 0.911]
	Comfort with AI	5	0.841	0.879 [0.859, 0.899]
	AI Technological Skills	4	0.807	0.782 [0.744, 0.821]
Work Engagement	Overall	13	0.957	0.750 [0.702, 0.798]

**Table 2 behavsci-15-01682-t002:** Goodness-of-fit Statistics for Competing Measurement Models.

Model	*χ* ^2^	*df*	*χ*^2^/*df*	CFI	TLI	RMSEA	SRMR	AIC	BIC
10-factor theoretical model	2874.785	1916	1.5	0.928	0.924	0.040 [0.037, 0.043]	0.0605	3202.785	3816.111
Single-factor model	6055.792	1941	3.12	0.69	0.678	0.083 [0.080, 0.085]	0.0961	6333.792	6853.623

Note. The single-factor model failed to converge (iteration limit reached), indicating it is an improper solution. Fit indices are reported from the last iteration but are not considered valid. SRMR was not applicable.

**Table 3 behavsci-15-01682-t003:** Means, Standard Deviations, and Intercorrelations Among Study Variables.

Variables	Mean	SD	1	2	3	4	5	6	7	8	9	10
1. CA	4.13	0.39	-									
2. Control	3.94	0.55	0.540 **	-								
3. Concern	4.2	0.55	0.832 **	0.281 **	-							
4. Curiosity	4.14	0.48	0.857 **	0.342 **	0.579 **	-						
5. Confidence	4.23	0.46	0.797 **	0.354 **	0.585 **	0.590 **	-					
6. JS1	3.53	1.14	0.388 **	0.218 **	0.310 **	0.356 **	0.332 **	-				
7. JS2	4.21	0.98	0.447 **	0.244 **	0.396 **	0.397 **	0.343 **	0.742 **	-			
8. SEAI1	5.63	0.6	0.383 **	0.172 **	0.297 **	0.255 **	0.348 **	0.351 **	0.349 **	-		
9. SEAI2	4.71	1.11	0.295 **	0.248 **	0.175 **	0.287 **	0.211 **	0.228 **	0.269 **	0.483 **	-	
10. AIA	2.52	0.87	−0.269 **	−0.151 **	−0.140 *	−0.237 **	−0.310 **	−0.391 **	−0.341 **	−0.358 **	−0.385 **	-
11. WE	4.72	1.12	0.570 **	0.339 **	0.457 **	0.520 **	0.446 **	0.692 **	0.749 **	0.393 **	0.291 **	−0.320 **

Note: * *p* < 0.05; ** *p* < 0.01.

**Table 4 behavsci-15-01682-t004:** (**a**). Regression Results for the Moderated Mediation Model. (**b**). Index of Moderated Mediation.

(**a**)							
**Outcome**	**Predictor**	** *b* **	** *SE* **	** *t* **	** *p* **	**LLCI**	**ULCI**
Model 1: Outcome = Extrinsic Satisfaction (JS1)/W = Comfort with AI (SEAI1)	CA (X)	−4.9042	2.1992	−2.2300	0.0265	−9.2320	−0.5764
	SEAI1 (W)	−5.1477	1.6639	−3.0938	0.0022	−8.4221	−1.8734
	AIA (Z)	−5.7176	2.4176	−2.3650	0.0187	−10.4753	−0.9600
	CA × SEAI1 (Int_1)	1.3880	0.4293	3.2333	0.0014	0.5432	2.2328
	CA × AIA (Int_2)	1.4051	0.6436	2.1833	0.0298	0.1386	2.6716
	SEAI1 × AIA (Int_3)	1.3401	0.5117	2.6188	0.0093	0.3331	2.3471
	CA × SEAI1 × AIA (Int_4)	−0.3509	0.1303	−2.6922	0.0075	−0.6074	−0.0944
Y = Work Engagement (WE)	CA (Direct Effect)	1.2912	0.1082	11.9354	0.0000	1.0783	1.5041
	JS1 (Mediator)	0.5400	0.0365	14.7740	0.0000	0.4680	0.6119
Model 2: Outcome = Extrinsic Satisfaction (JS1)/W = AI Technological Skills (SEAI2)	CA (X)	−3.5570	2.2347	−1.5917	0.1124	−7.9580	0.8439
	SEAI2 (W)	−5.5428	1.6851	−3.2892	0.0011	−8.8590	−2.2266
	AIA (Z)	−4.1837	1.8214	−2.2970	0.0223	−7.7680	−0.5994
	CA × SEAI2 (Int_1)	1.5080	0.4319	3.4917	0.0006	0.6581	2.3580
	CA × AIA (Int_2)	1.0282	0.4752	2.1636	0.0313	0.0930	1.9634
	SEAI2 × AIA (Int_3)	1.3196	0.5678	2.3242	0.0208	0.2023	2.4369
	CA × SEAI2 × AIA (Int_4)	−0.3502	0.1462	−2.3958	0.0172	−0.6379	−0.0626
Y = Work Engagement (WE)	CA (Direct Effect)	1.2912	0.1082	11.9354	0.0000	1.0783	1.5041
	JS1 (Mediator)	0.5400	0.0365	14.7740	0.0000	0.4680	0.6119
Model 3: Outcome = Intrinsic Satisfaction (JS2)/W = Comfort with AI (SEAI1)	SEAI1 (W)	−2.9548	1.7984	−1.6430	0.1014	−6.4938	0.5841
	AIA (Z)	−3.8508	1.3600	−2.8315	0.0049	−6.5270	−1.1746
	CA × SEAI1 (Int_1)	−3.7399	1.9751	−1.8935	0.0592	−7.6265	0.1467
	CA × AIA (Int_2)	1.0223	0.3511	2.9115	0.0039	0.3313	1.7132
	SEAI1 × AIA (Int_3)	0.7756	0.5265	1.4732	0.1417	−0.2604	1.8117
	CA × SEAI1 × AIA (Int_4)	0.9571	0.4189	2.2846	0.0230	0.1327	1.7815
	SEAI1 (W)	−0.2171	0.1067	−2.0341	0.0428	−0.4272	−0.0071
Y = Work Engagement (WE)	JS2 (Mediator)	0.7392	0.0905	8.1725	0.0000	0.5613	0.9172
	JS2 (Mediator)	0.8684	0.0344	25.2323	0.0000	0.8007	0.9361
(**b**)							
**Model**	**Index**	**Effect Size**		**Boot SE**		**Boot LLCI**	**Boot ULCI**
Model 1 (M = JS1, W = SEAI1)	Overall Index (X × W × Z)	−0.1895 *		0.0919		−0.4243	−0.0584
	Conditional Index by AIA (Z)						
	Low AI anxiety	0.6083 *		0.2286		0.2407	1.1574
	Moderate AI anxiety	0.4654 *		0.1772		0.1737	0.8775
	High AI anxiety	0.3225 *		0.1419		0.0602	0.6212
Model 2 (M = JS1, W = SEAI2)	Overall Index (X × W × Z)	−0.1891 *		0.0849		−0.3901	−0.0575
	Conditional Index by AIA (Z)						
	Low AI anxiety	0.6734 *		0.1988		0.3097	1.0930
	Moderate AI anxiety	0.5307 *		0.1591		0.2263	0.8559
	High AI anxiety	0.3881 *		0.1388		0.1050	0.6476
Model 3 (M = JS2, W = SEAI1)	Overall Index (X × W × Z)	−0.1886 *		0.1248		−0.5106	−0.0156
	Conditional Index by AIA (Z)						
	Low AI anxiety	0.7472 *		0.3158		0.2825	1.5341
	Moderate AI anxiety	0.6050 *		0.2425		0.2342	1.1938
	High AI anxiety	0.4628 *		0.1888		0.1361	0.8738

Note. All predictor and outcome variables were standardized (Z-scored) prior to analysis. *b* represents the unstandardized regression coefficient. *SE* = standard error; LLCI and ULCI = lower and upper bounds of the 95% confidence interval. CA = Career Adaptability; JS1 = Extrinsic Satisfaction; JS2 = Intrinsic Satisfaction; SEAI1 = Comfort with AI; SEAI2 = AI Technological Skills; AIA = AI Anxiety; WE = Work Engagement. X (The independent variable): CA (all models). W (Primary Moderator): The first moderator, SEAI1 (Models 1 and 3) or SEAI2 (Model 2). It moderates the path from X to the Mediator. Z (Second-Stage Moderator): The second moderator, AIA (all models). It moderates the X × W interaction. Int_1: The two-way interaction term for X × W (e.g., CA × SEAI1). Int_2: The two-way interaction term for X × Z (e.g., CA × AIA). Int_3: The two-way interaction term for W × Z (e.g., SEAI1 × AIA). Int_4: The three-way interaction term for X × W × Z (e.g., CA × SEAI1 × AIA). Mediator (Path b): The regression coefficient for the mediator (JS1 or JS2) predicting the final outcome (WE). Direct Effect (Path c′): The regression coefficient for the focal predictor (CA) predicting the final outcome (WE), controlling for the mediator. Index = Index of Moderated Moderated Mediation. Effect size = unstandardized conditional index. Boot SE = Bootstrap Standard Error. Boot LLCI and Boot ULCI = Bootstrap 95% bias-corrected confidence intervals. Low, moderate, and high levels of the moderator (AI Anxiety) correspond to ±1 SD from the mean and the mean. * denotes a 95% confidence interval that does not contain zero.

**Table 5 behavsci-15-01682-t005:** (**a**) Conditional Indirect Effects (Model 1: X = CA4, M = JS2, W = SEAI2). (**b**) Conditional Indirect Effects (Model 2: X = CA4, M = JS1, W = SEAI2). (**c**) Conditional Indirect Effects (Model 3: X = CA1, M = JS2, W = SEAI1).

(**a**)					
**AI Technological Skills (W)**	**AIA (Z)**	**Effect Size**	**Boot SE**	**Boot LLCI**	**Boot ULCI**
Low	Low	0.5183	0.2689	−0.0459	1.0109
Low	Middle	0.4795 *	0.1696	0.1231	0.7947
Low	High	0.4407 *	0.1362	0.1674	0.7037
Middle	Low	0.9276 *	0.2245	0.4946	1.3712
Middle	Middle	0.7927 *	0.1466	0.5132	1.0723
Middle	High	0.6578 *	0.1594	0.3409	0.9640
High	Low	1.3369 *	0.2752	0.8248	1.9035
High	Middle	1.1058 *	0.2066	0.7278	1.5362
High	High	0.8748 *	0.2247	0.4218	1.3084
(**b**)					
**AI Technological Skills (W)**	**AIA (Z)**	**Effect Size**	**Boot SE**	**Boot LLCI**	**Boot ULCI**
Low	Low	0.3288	0.2515	−0.1782	0.8187
Low	Middle	0.3587 *	0.1598	0.0355	0.6733
Low	High	0.3885 *	0.1172	0.1567	0.6202
Middle	Low	0.8461 *	0.1843	0.4992	1.2184
Middle	Middle	0.7664 *	0.1244	0.5261	1.0136
Middle	High	0.6867 *	0.1530	0.3687	0.9723
High	Low	1.3633 *	0.2266	0.9249	1.8192
High	Middle	1.1740 *	0.1878	0.8095	1.5412
High	High	0.9848 *	0.2362	0.4884	1.4150
(**c**)					
**Comfort with AI (W)**	**AIA (Z)**	**Effect Size**	**Boot SE**	**Boot LLCI**	**Boot ULCI**
Low	Low	1.1310 *	0.3869	0.3083	1.8057
Low	Middle	1.0309 *	0.2466	0.5054	1.4783
Low	High	0.9309 *	0.2028	0.5262	1.3202
Middle	Low	1.6338 *	0.3128	1.0170	2.2249
Middle	Middle	1.4380 *	0.2036	1.0286	1.8264
Middle	High	1.2423 *	0.2245	0.7782	1.6636
High	Low	2.1365 *	0.3691	1.4226	2.8684
High	Middle	1.8451 *	0.2745	1.3235	2.4020
High	High	1.5537 *	0.3032	0.9310	2.1307

Note. Indirect effects of Career Adaptability (CA) on Work Engagement (WE) via Extrinsic Satisfaction (JS1), with Comfort with AI (SEAI1) as the moderator (W) (Models a and b), or via Intrinsic Satisfaction (JS2), with Comfort with AI (SEAI1) as the moderator (W) (Model c). Effects are estimated from models that control for covariates (Models a and b), or without covariates (Model c). Bootstrapped CIs (5000 samples) are bias-corrected. Low, moderate, and high levels of the moderators (W and Z) correspond to ±1 SD from the mean and the mean. * denotes a 95% confidence interval that does not contain zero. CA = Career Adaptability; JS1 = Extrinsic Satisfaction; JS2 = Intrinsic Satisfaction; SEAI1 = Comfort with AI; SEAI2 = AI Technological Skills; WE = Work Engagement. LLCI and ULCI = lower and upper bounds of the 95% confidence interval. *SE* = standard error.

**Table 6 behavsci-15-01682-t006:** Unstandardized Path Coefficients from Latent SEM Model.

Path	Predictor	Outcome	*b*	BCa 95% CI
a1	CA (2nd Order)	JS1	1.249 ***	[0.901, 1.885]
a2	CA (2nd Order)	JS2	1.361 ***	[1.023, 1.967]
b1	JS1	WE	0.107	[−0.030, 0.250]
b2	JS2	WE	0.716 ***	[0.548, 0.912]
c	CA (2nd Order)	WE	0.715 ***	[0.385, 1.396]

Note. *b* = unstandardized regression coefficient. BCa 95% CI = Bias-corrected 95% Confidence Interval based on 5000 bootstrap samples. Paths are significant if the 95% CI does not cross zero. CA = Career Adaptability (Second-Order Factor); JS1 = Extrinsic Job Satisfaction; JS2 = Intrinsic Job Satisfaction; WE = Work Engagement. *** *p* < 0.001.

**Table 7 behavsci-15-01682-t007:** Total Indirect Effects from Latent SEM Model.

Path	Total Indirect Effect (*b*)	BCa 95% CI	Bootstrap *p*-Value
CA → JS1, JS2 → WE	1.109 ***	[0.796, 1.598]	0.000

Note. *b* = unstandardized total indirect effect. BCa 95% CI = Bias-corrected 95% Confidence Interval based on 5000 bootstrap samples. Effects are significant if the 95% CI does not cross zero. Total indirect effects represent the sum of effects channeled through both JS1 and JS2. CA = Career Adaptability (Second-Order Factor); JS1 = Extrinsic Job Satisfaction; JS2 = Intrinsic Job Satisfaction; WE = Work Engagement. *** *p* < 0.001.

## Data Availability

The original contributions presented in this study are included in the article. Further inquiries can be directed to the corresponding author.

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
