# Peer review of "The Relationship Between Career Adaptability and Work Engagement Among Young Chinese Workers: Mediating Role of Job Satisfaction and Moderating Effects of Artificial Intelligence Self-Efficacy and Anxiety"

_behavsci, 2025, doi:10.3390/bs15121682_

Round 1

Reviewer 1 Report

Comments and Suggestions for Authors

Dear authors,
Congratulations on a very original and relevant topic for today.
I would just like to make a couple of suggestions to improve the transparency and strength of the article.
Title - I would suggest a more concise title. The length makes it hard for the reader to engage with the article.

Introduction

There are too many citations for the same statement in the introduction. For example, “The rapid advancement of Artificial Intelligence (AI) is profoundly transforming the workplace (Schwab, 2016; Acemoglu et al., 2022; Brynjolfsson & McAfee, 2017; Schwab, 2024.” Are four references really necessary for one sentence? The excess makes it difficult for the reader to read the text fluently. 
The first paragraph repeats ideas (lines 43-45 x lines 46-48). I suggest integrating them.
Some references could be updated to include studies from the last five years. 
The theoretical background only presents the concepts as a list, without presenting a problematized relationship between them to justify the choice of such variables. The chosen format makes it difficult to quickly identify the gap in the literature. Reformulation is needed to present the gap, which so far seems uncertain.

Lines 84-87: hw does a reference from 1997 apply to a statement about IA? If you want to cite Bandura, you need to elaborate on how his theory explains the situation. This adjustment is necessary for the reader to understand the need for H2 and H3.
The introduction does not explain why the sample consists of young workers. Why this selection?
NO 1.2 in the title has “research questions,” but they are only presented in 1.3.
1.4 seems repetitive and unnecessary since these ideas are in the discussion.

Method
In the method, the study design seems to be mislabeled. They call it an experiment when it was actually a cross-sectional survey. 
In lines 159-160, they present a data point called RMB. I suggest inserting a note explaining what it is so that the reader knows what it refers to.
Line 166: Which compensation? Was it equal among all participants? What were the inclusion criteria, and how did you gather the participants?
Line 170: Was the exploratory and confirmatory analysis performed with the same sample? This is not recommended.
Please provide the validity and reliability data for the original versions of the scales used.
Regarding data cleaning, there is information that cases were excluded due to errors or unreliable responses, but there is no information on the rules used to apply this.
The inclusion and exclusion criteria are unclear.

Why was ML used for Likert scales? Why not a robust estimator such as MLR?
Results
Such a strong focus on the validation of the instruments detracts from what should be the focus of the article. The article reads more like a validation study than a mediation study. 
Regarding reliability, if possible, I suggest reporting the omega.
The three-way effect is only marginally significant (p ≈ 0.049) and the overall moderate mediation index is not significant, which requires qualifying H3 as partially supported. The biggest error, however, is in the narrative: the text states that indirect effects cease to be significant under high anxiety in AI, but Table 5 shows exactly the opposite, with significant effects in all conditions, including this one. Despite this, the authors declare all hypotheses as supported. Important information is still missing, such as R², ΔR², slope graphs, and the explanation of indirect and total effects, which are essential for a complete interpretation of the results.
Discussion

I found the discussion quite difficult to read due to the excessive use of abbreviations. While the introduction seemed to have too many citations, here there are many sentences that need support from the literature, which is lacking.
The language of the discussion should be softened, as it may seem exaggerated or incoherent in light of the results.

For example, the text states that, under high anxiety in AI, the indirect effects cease to be significant, but this is not true according to Table 5. It also states that hypothesis H3 was confirmed, although the overall moderate mediation index is not significant and the three-way effect is only marginal.
Furthermore, it uses language that implies causality in a study that cannot infer causality.
In addition, decisions made after seeing the data (such as using only the technical competence subscale or total satisfaction) are not presented as exploratory. The discussion also exaggerates the importance of the measurement results and treats common method bias as resolved with a weak test. Finally, the practical implications are too broad for the sample and study design, and the limitations are not discussed critically enough.
Limitations
The text already mentions important limitations; however, for greater transparency, it is necessary to mention the use of the same sample for exploratory and confirmatory analysis.
The measures were re-specified and need justification, such as:
CA was expanded, 61→33 items, with 5 factors including OCB/collective efficacy, and a non-significant 2nd order load; this alters the construct and should be assumed as a limitation. JSS reduced from 36→22 items and forced to 2 factors, with error correlations released, which limits comparability with the original instrument. 
Recognize possible panel and self-selection sample biases, and the absence of weighting/stratification.

Reviewer 2 Report

Comments and Suggestions for Authors

The study examines a moderated mediation model in which career adaptability influences work engagement via job satisfaction, with AI self-efficacy and AI anxiety as moderators, using a young worker sample. The theoretical framing (COR/SCT) is appropriate and the main paths (CA→JS; JS→WE) are well supported. However, several issues require revision:

  1. Measurement validity and overfitting risk. Multiple scales were pruned post-hoc and residual covariances were added to improve fit. Running EFA and CFA on the same dataset, coupled with post-hoc re-specification, increases overfitting risk. Please provide a full item map (retained/removed), factor loadings/cross-loadings, and justification for residual covariances; ideally, confirm the structure in a second subsample (split-sample) or a follow-up study.

  2. Modeling strategy. You validated measures via CFA but tested the moderated mediation with observed composites in PROCESS. This ignores measurement error and can bias effects. Please re-estimate the key model using latent SEM with latent interactions (or, at minimum, add a robustness check using a latent variable approach).

  3. Claims vs. evidence. The “double-edged sword” narrative should be moderated: the overall index of moderated mediation is not significant. Re-write the abstract and discussion to reflect where conditional indirect effects are significant, and add plots of conditional effects with confidence intervals.

  4. Discriminant validity and consistency of work engagement. Very high correlations between WE dimensions suggest limited discriminant validity. Choose a consistent strategy (e.g., use a total WE score in main analyses) or explicitly justify the multidimensional approach (e.g., HTMT, alternative models).

  5. Common-method bias and design limitations. Beyond Harman’s one-factor test, consider a latent method factor in CFA or marker variable approach, and acknowledge cross-sectional, self-report, online-panel constraints more explicitly. Report attention checks, exclusion criteria, and compensation details.

  6. Reporting additions. Include: (i) full bootstrap CIs for conditional indirect effects, (ii) standardized effect sizes (e.g., ΔR² for interaction terms), VIF/tolerance diagnostics, and (iii) figures of simple slopes/interactions.

  7. Data availability and reproducibility. Share anonymized data, code (syntax for PROCESS/SEM), and full materials in OSF/Zenodo to enhance transparency.

  8. Language and terminology. Tighten long sentences, remove redundancies, and standardize terminology (e.g., “vigor” vs. “vitality”; consistent labels for AI constructs).

Overall, the manuscript addresses a timely question with promising results, but it requires a major revision focused on measurement validity, appropriate latent modeling, and tighter alignment of claims with statistical evidence.

Round 2

Reviewer 1 Report

Comments and Suggestions for Authors

The authors have addressed the previous comments thoroughly and have significantly improved the manuscript. I believe the paper is now suitable for publication in its current form.

Author Response

Thank you for your comments.

Reviewer 2 Report

Comments and Suggestions for Authors

The manuscript has improved substantially and now makes a clear and relevant contribution to the literature on career adaptability and work engagement among young workers in China. The theoretical positioning is clear, the measurement validation is adequately reported, and the moderated-mediation logic is consistently carried through the Results and Discussion.

Only minor items remain, which are largely editorial and will strengthen clarity:

  1. Align the take-home message across sections. Emphasize early (Abstract/Introduction) and again in the Discussion that intrinsic job satisfaction is the key mediating pathway, while AI self-efficacy and AI anxiety condition selected relationships. Keep this narrative consistent with the statistical evidence you report.

  2. Streamline small redundancies. A few sentences in the Introduction repeat prior points; trimming them will tighten the flow without removing content.

  3. Terminology and acronyms. Please ensure consistent use of “Artificial Intelligence (AI)” (avoid variants such as “Artificial Technology”) and uniform use of scale acronyms (e.g., UWES) at first mention.

  4. Limitations and generalizability. You already acknowledge the cross-sectional, self-report design. Add one line clarifying the implications for causal inference and suggest a simple longitudinal or multi-source design for future work.

  5. Figures/Tables housekeeping. Check that figure captions, panel labels, and in-text cross-references (e.g., Fig. 1a–c; Tables 2–4) are fully consistent.

With these minor edits, I recommend acceptance after minor revision. Congratulations on a solid and relevant piece of work.

Comments on the Quality of English Language

The English is overall good and fully understandable. A light copy-edit would help with precision and flow. In particular:

  • Use “Artificial Intelligence (AI)” consistently; avoid alternative phrasings (e.g., “Artificial Technology”).

  • Standardize acronyms (e.g., UWES) and define each at first mention.

  • Break a few long sentences in the Introduction to improve readability.
    These are stylistic refinements; they do not affect the substance of the paper.
